# What Should Embeddings Embed? Autoregressive Models Represent Latent Generating Distributions

**Liyi Zhang**                                                        *zhang.liyi@princeton.edu*
*Department of Computer Science*
*Princeton University*

**Michael Y. Li**                                                    *michaelyli@stanford.edu*
*Department of Computer Science*
*Stanford University*

**R. Thomas McCoy**                                                  *tom.mccoy@yale.edu*
*Department of Linguistics and Wu Tsai Institute*
*Yale University*

**Theodore R. Sumers**                                               *ted@anthropic.com*
*Anthropic*

**Jian-Qiao Zhu**                                                    *jz5204@princeton.edu*
*Department of Computer Science*
*Princeton University*

**Thomas L. Griffiths**                                              *tomg@princeton.edu*
*Departments of Psychology and Computer Science*
*Princeton University*

**Reviewed on OpenReview:** *https://openreview.net/forum?id=YyMACp98Kz*

## Abstract

Autoregressive language models have demonstrated a remarkable ability to extract latent structure from text. The embeddings from large language models have been shown to capture aspects of the syntax and semantics of language. But what *should* embeddings represent? We show that the embeddings from autoregressive models correspond to predictive sufficient statistics. By identifying settings where the predictive sufficient statistics are interpretable distributions over latent variables, including exchangeable models and latent state models, we show that embeddings of autoregressive models encode these explainable quantities of interest. We conduct empirical probing studies to extract information from transformers about latent generating distributions. Furthermore, we show that these embeddings generalize to out-of-distribution cases, do not exhibit token memorization, and that the information we identify is more easily recovered than other related measures. Next, we extend our analysis of exchangeable models to more realistic scenarios where the predictive sufficient statistic is difficult to identify by focusing on an interpretable subcomponent of language, topics. We show that large language models encode topic mixtures inferred by latent Dirichlet allocation (LDA) in both synthetic datasets and natural corpora.

## 1 Introduction

Autoregressive language models (LMs) are trained to predict the next token in a sequence (e.g., Bengio et al., 2000). Many large language models (LLMs) use the autoregressive objective for pretraining (e.g., Radford et al., 2019), and their document-level embeddings have been shown to capture elements of latent structure

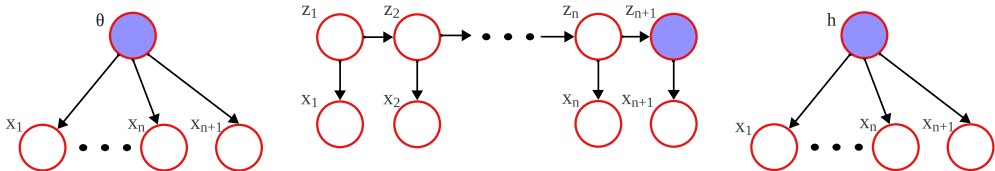

Figure 1: Three data generation processes where prediction of the next token $x_{n+1}$ is independent from previous tokens $x_{1:n}$ given a predictive sufficient statistic. The left corresponds to exchangeable data, the middle to latent state models, and the right to discrete hypotheses. The relevant predictive sufficient statistics are the sufficient statistic for $\theta, z_{n+1}, h$ (or $p(\theta|x_{1:n})$, $p(z_{n+1}|x_{1:n})$, and $p(h|x_{1:n})$ respectively). We show the embeddings learned by autoregressive transformers represent this information.

that appear in text, such as agent properties (Andreas, 2022) and syntax (Hewitt & Manning, 2019). While it is intuitive that representing this information assists in next token prediction, we lack a formal understanding of which aspects of text should be represented and why. Given that these embeddings are often used for downstream tasks such as sequence classification and sentiment analysis, understanding what they represent and why is key to interpreting and building on these models.

Analyzing the representations formed by LLMs is challenging partly due to polysemanticity, where a neuron may activate for several distinct concepts (Cunningham et al., 2023). Previous work has developed methods for probing LLM internal representations for specific concepts (Li et al., 2023; Meng et al., 2022; Zheng et al., 2023; Tenney et al., 2019). These efforts can be guided more effectively by developing a general theory of what aspects of the data embeddings should represent.

In this work, we investigate several cases where the representations of autoregressive LMs can be formally connected with those of a Bayes-optimal agent. Because of their expressive architecture and their objective, LM embeddings should encode latent structure such that the next word $x_{n+1}$ is independent from previous words $x_{1:n}$ when conditioned on that structure. This corresponds to the notion of a *predictive sufficient statistic*. Using this connection, we show that optimal content of embeddings can be identified in two major cases: 1) independent identically distributed data, where the embedding should capture the sufficient statistics of the data; 2) latent state models, where the embedding should encode the posterior distribution over the next latent state given the data. An application of case 1 that we explore is topic models based on latent Dirichlet allocation (LDA), where the LM embedding encodes the topic mixtures of a given document. We use probing to confirm that the relevant information can be decoded from LM embeddings, and that content that is not expected to be captured via predictive sufficient statistics is more challenging to recover.

Next, we extend our analysis of exchangeable models to more realistic scenarios by generating synthetic data where only a fraction of words are generated from LDA. We confirm that language models represent topic distributions when a sufficiently large portion of the text serves this semantic component. We finally show that the encodings of pretrained LLMs (GPT-2, LLAMA 2, and BERT) contain information analogous to that extracted by LDA from two natural corpora, supporting the hypothesis that LLMs implicitly perform Bayesian inference in the naturalistic setting.

Our analysis provides concrete examples where we can analytically identify what embeddings should represent, and empirical confirmation that this information is indeed captured in models trained with an autoregressive objective. By linking LM representations to Bayesian inference, our approach supports recent arguments that the behavior of LMs can be interpreted through comparison with Bayes-optimal agents (Mikulik et al., 2020). Bayesian inference encourages the agent to summarize data by constructing a generative process. Thus, understanding this connection can help LLM researchers hypothesize what features are encoded in LLM internals as done in *mechanistic interpretability* (Cunningham et al., 2023); by recovering latent generating distributions from LM, we show that LMs not only learn concepts, but also learn their uncertainty representation, complementing recent studies at the behavioral level (Gruver et al., 2024).

## 2    Related work

**Embedding analysis in language models**   The embeddings produced by language models have been investigated in detail (Gupta et al., 2015; Köhn, 2015; Ettinger et al., 2016; Adi et al., 2017; Hupkes et al., 2018); for reviews, see Rogers et al. (2020) and Belinkov (2022). They have been shown to capture aspects of the latent structure of text, including part of speech (Shi et al., 2016; Belinkov et al., 2017), sentence structure (Tenney et al., 2019; Hewitt & Manning, 2019; Liu et al., 2019; Lin et al., 2019), sentiment (Radford et al., 2017), semantic roles (Ettinger et al., 2016; Tenney et al., 2019), and agent properties (Andreas, 2022). However, our work is motivated from a fundamentally different perspective. Instead of focusing on *what* is captured in the embeddings of these models, we provide a general theory of language models as Bayes-optimal agents to explain *why* these kinds of structure might be represented due to statistical properties of the training data. Characterizing the ideal representations of a Bayes-optimal agent complements recent efforts in *mechanistic interpretability* (Nanda et al., 2023; Cunningham et al., 2023; Cammarata et al., 2020) by identifying optimal representations of data; this can inform the search for interpretable features that are used in LLMs' circuit-level computations.

**Implicit Bayesian inference**   Several previous papers have analyzed LLMs by making a connection to Bayesian inference. Of these, Xie et al. (2021), McCoy et al. (2023), and Wang et al. (2024b) analyze the in-context learning behavior of LLMs. However, we study what models should encode based on the autoregressive objective that is typically used to train LLMs. Zheng et al. (2023) focus on topic models embedded in LSTMs, while we extend the connection to more general cases. Wang et al. (2024a) provides a novel method to extract from language models concepts that come in the form of a mixed membership model (including the topic model). Our focus is more general and explanatory as we analyze why concepts that come in the form of sufficient stats should be embedded by autoregressive language models.

Belief state inference (BSI) on latent state models (Shai et al., 2024) decodes the distributions of the current latent state from the transformer embeddings on observed sequences. Our more general theory shows that this ability comes from the embedding encoding predictive sufficient statistics for the sequence, which more directly relates to the distribution of the next latent state given observations. Concurrently, Ye et al. (2024) also used de Finetti's theorem to support probabilistic reasoning in transformers, but focused on *in-context learning* in the context of Bayesian linear regression. We analyze transformer internal representations and identify optimal embeddings in data that violate the exchangeability assumption.

**Theory**   Tishby & Zaslavsky (2015) gave a theoretical analysis of optimal embeddings in deep networks from an information-theoretic perspective. We take a complementary Bayesian approach, explicitly connecting optimal embeddings to predictive sufficient statistics and presenting extensive experiments confirming that the predicted content of embeddings is tracked by transformer-based LMs. Metalearned RNNs have also been shown to encode information equivalent to a Bayesian posterior distribution (Mikulik et al., 2020). Furthermore, recent work has also demonstrated that transformers behave like the Bayes-optimal predictor in linear regression settings (Panwar et al., 2024; Garg et al., 2022; Akyürek et al., 2023) and can approximate the posterior predictive distributions of probabilistic models such as Gaussian processes and Bayesian neural networks (Müller et al., 2022). We extend this analysis to general autoregressive language models and consider more general generative processes and what posterior distributions they should capture in these cases.

## 3    Optimal embeddings

Assume we have a sequence $x_{1:n}$ and an autoregressive language model that predicts the next item in the sequence, $p(x_{n+1}|x_{1:n})$. We denote the LM embedding for sequence $x_{1:n}$ as $\phi_n = f(x_{1:n})$. The distribution $p(x_{n+1}|x_{1:n})$ is a function $g(\phi_n)$ of this embedding, with that function implemented by the final layer of the neural network instantiating the LM such that the probability of the next element $x_{n+1}$ only depends on $\phi_n$. This establishes our question: what should $\phi_n$ represent in order to accurately predict $x_{n+1}$?

The key idea behind our approach is that we can identify situations where $\phi_n$ contains all of the information from $x_{1:n}$ required to predict $x_{n+1}$ by using the notion of a *sufficient statistic* (Gelman et al., 2004). Given a distribution $p(x)$ with parameters $\theta$, a statistic $s(x)$ is sufficient for $\theta$ if the conditional distribution of $x$

given $s$ does not depend on $\theta$. In other words, if we only know $s$, we can estimate $\theta$ just as well as if we know $x$. In the autoregressive setting, we care about *predictive sufficiency* (Bernardo & Smith, 2000). A statistic $s(x_{1:n})$ is predictive sufficient for the sequence $x_{1:n}$ if

$$p(x_{n+1}|x_{1:n}) = p(x_{n+1}|s(x_{1:n})). \tag{1}$$

Crucially, if a model performs autoregressive modeling perfectly, it learns a predictive sufficient statistic.

In the remainder of this section we describe two cases where predictive sufficient statistics can be easily identified and could plausibly be represented by a neural network (Figure 1). These cases correspond to common applications of machine learning models as well as classic models for text such as hidden Markov models (Jurafsky & Martin, 2008). First, when $x_{1:n}$ are independently sampled conditioned on an unknown parameter, $\phi_n$ needs only represent the sufficient statistic of this sequence. Second, when $x_{1:n}$ are generated by a state space model (in the discrete case, a hidden Markov model), $\phi_n$ needs only represent the posterior distribution over the next latent state given $x_{1:n}$. In each case we explain how $p(x_{n+1}|x_{1:n})$ factorizes to make it possible for $x_{1:n}$ to be summarized by some $\phi_n$ and identify the form of the corresponding $g(\phi_n)$.

### 3.1 Case 1: Exchangeable models

Predictive sufficiency is straightforward to establish in *exchangeable* models, where the probability of a sequence remains the same under permutation of the order of its elements. That is, a sequence is exchangeable if $p(x_{1:N}) = p(x_{\pi(1:N)})$ for some permutation $\pi$. Any exchangeable model can be re-expressed in terms of the $x_i$ being sampled independently and identically distributed according to a latent distribution $p(x|\theta)$ parameterized by $\theta$, with $p(x_{1:N}) = \int_\theta \prod_i p(x_i|\theta)p(\theta)\,d\theta$ (Gelman et al., 2004). This idea leads to the following proposition:

*Proposition.* Given an exchangeable sequence $x_{1:N}$ where each $x_i$ is of dimension $d_x$, and given functions $f : \mathbb{R}^{nd_x} \mapsto \mathbb{R}^{d_m}$, $g : \mathbb{R}^{d_m} \mapsto \mathbb{R}^{d_x}$ such that, for each $1 \le n \le N$, $g \circ f(x_{1:n}) = p(x_{n+1}|x_{1:n})\,\forall x_{n+1}$, $f(x_{1:n})$ is a sufficient statistic for $x_{1:n}$.

In other words, if we have a perfect autoregressive predictor that is composable into $g \circ f$, the output of $f$ is a sufficient statistic for its sequence input.

*Proof.* The result follows from the fact that for exchangeable sequences, general sufficiency is equivalent to predictive sufficiency (Bernardo & Smith, 2000). Because $p(x_{n+1}|x_{1:n}) = g(f(x_{1:n}))\,\forall n$, $f(x_{1:n})$ is a predictive sufficient statistic for the sequence, and it is also a sufficient statistic. $\square$

The resulting sufficient statistic also fully specifies the posterior on the parameters of the generating distribution, $p(\theta|x_{1:n})$. Sufficient statistics are easily identified for a wide range of distributions, including all exponential family distributions (Bernardo & Smith, 2000), and are easy to represent. Since the LM's predictive distribution decomposes into the form above, this result gives us strong predictions about the contents of embeddings for models trained on exchangeable data.

#### 3.1.1 Case 1.1: Discrete hypothesis spaces

In a more specific version of Case 1, assume each $x_i$ is generated independently from some unknown generative model, and let $\mathcal{H}$ denote the discrete set of hypotheses $h$ about the identity of this model. In this case, $x_{1:n}$ are exchangeable, but any sufficient statistics might be difficult to identify. The posterior predictive distribution can be written as

$$p(x_{n+1}|x_{1:n}) = \sum_{h \in \mathcal{H}} p(x_{n+1}|h)p(h|x_{1:n}).$$

By an argument similar to Case 2, $p(h|x_{1:n})$ is a predictive sufficient statistic in this model and an embedding thus need only capture this posterior distribution.

### 3.1.2 Case 1.2: Topic Models

Latent Dirichlet Allocation (LDA; Blei et al. (2001)) is an exchangeable generative model that is widely used for modelling the topic structure of documents. A document is generated from a mixture of $K$ topics. Each topic is a distribution over the vocabulary; e.g., a topic corresponding to geology might assign high probability to words such as *mineral* or *sedimentary*. We detail the full generative process in Appendix A.2.

After LDA is trained on a corpus, the inferred quantities can be used to explore the corpus. We denote by $\theta_i$ as the latent variable that stands for each document's underlying topic mixture. Because topic mixture $\theta$ and word distribution $\beta$ form the sufficient statistic, our theory suggests that autoregressive LMs should implicitly encode the topic structure of a document.

## 3.2 Case 2: Latent state models

In a latent state model, each $x_i$ is generated based on a latent variable $z_i$. These $z_i$ are interdependent, with $z_i$ being generated from a distribution conditioned on $x_{i-1}$. Common latent state models include Kalman filters (where $x_i$ and $z_i$ are continuous and the emission and transition functions, $p(x_i|z_i)$ and $p(z_i|z_{i-1})$, are linear-Gaussian) (Kalman, 1960) and hidden Markov models (where the $z_i$ are discrete) (Baum & Petrie, 1966). In a latent state model, the posterior predictive distribution is

$$p(x_{n+1}|x_{1:n}) = \int p(x_{n+1}|z_{n+1})p(z_{n+1}|x_{1:n}) \, dz_{n+1}. \tag{2}$$

In this case, the posterior over the next latent state $p(z_{n+1}|x_{1:n})$ captures all of the information in $x_{1:n}$ relevant to predicting $x_{n+1}$, rendering $x_{n+1}$ independent of $x_{1:n}$ when conditioned on this statistic. If we were to drop conditional independence assumptions in a latent state model, one would need $p(x_{n+1}|z_{n+1}, x_{1:n})$ instead of $p(x_{n+1}|z_{n+1})$ for the equality. Thus, $p(z_{n+1}|x_{1:n})$ is a predictive sufficient statistic in this model. However, there is another predictive sufficient statistic. We can write

$$p(x_{n+1}|x_{1:n}) = \int p(x_{n+1}|z_n)p(z_n|x_{1:n}) \, dz_n, \tag{3}$$

showing $p(z_n|x_{1:n})$ to also be a predictive sufficient statistic. Which of these will be favored will depend on how easily the relevant integrals can be approximated.

More formally, we consider an embedding a representation $\phi_n$ such that a fixed operator $g(\phi_n)$ can produce $p(x_{n+1}|x_{1:n})$. The distributions $p(z_{n+1}|x_{1:n})$ and $p(z_n|x_{1:n})$ satisfy this characterization, with Equations 2 and 3 showing that the relevant operator is the integral of $p(x_{n+1}|z_{n+1})$ over $z_{n+1}$ or $p(x_{n+1}|z_n)$ over $z_n$ and $z_{n+1}$. That operator can be easily approximated linearly and hence by a single layer of a neural network. The embedding $\phi_n$ thus needs only represent $p(z_{n+1}|x_{1:n})$ or $p(z_n|x_{1:n})$, depending on the relative ease of approximating the relevant integrals in a specific autoregressive predictor.

## 3.3 Probing embeddings to recover predictive sufficient statistics

These cases specify information that should be encoded in neural network embeddings. This sets up the second element of our approach, which is building probes to check this hypothesis. We focus on transformers (Vaswani et al., 2017), since they are widely used as language models. We denote the decoding target for each sequence $x_{1:n}$ by a vector $t_n$. Given a trained transformer, we decode $t_n$ by training a second model (a probe) to predict the target $t_n$ from the embedding $\phi_n$ of the corresponding document $x_{1:n}$. The embedding is defined to be the last layer embedding, which is what researchers and practitioners typically use as a document representation. The probe $g$ maps from the sequence embedding to the target. To ensure that the relevant statistical information is contained in the LM, not in the probe, we keep the probe simple by defining it as a linear layer with softmax activations: $g(\phi_n) = \text{Softmax}(\text{Linear}(\phi_n))$. In each case, the probe uses the last-layer, last-token embedding as its input, unless otherwise specified.

Table 1: Probing results on discrete hypothesis spaces. In general, the probe achieves strong performance in recovering a 784-length vector, and performance increases as task difficulty decreases. Average and standard error across 10 random seeds are reported.

| | Equal Width | | Unequal Width | |
| --- | --- | --- | --- | --- |
| Sample Size | Accuracy ↑ | Squared Loss ↓ | Accuracy ↑ | Squared Loss ↓ |
| 20 | $87.3 \pm 0.4\%$ | $0.173 \pm 0.005$ | $66 \pm 0.5\%$ | $0.29 \pm 0.004$ |
| 50 | $99.5 \pm 0.1\%$ | $0.008 \pm 0.001$ | $88.5 \pm 0.3\%$ | $0.159 \pm 0.003$ |

## 4 Empirical analysis

We have identified cases where a predictive sufficient statistic is expected to be encoded by an autoregressive model. In this section, we use probing analyses on transformers to empirically validate this hypothesis. Ablation studies on memorization, parsimonious properties of embeddings, and decoding alternative quantities further support this idea. We use the Adam optimizer (Kingma & Ba, 2014) in all experiments. Code is available at github.com/zhang-liyi/llm-embeddings.

### 4.1 Simple exchangeable models

#### 4.1.1 Generative model

**Bayesian conjugate models** We start with three data generating distributions: Gaussian-Gamma, Beta-Bernoulli, and Gamma-Exponential models.

For $M$ Gaussian-Gamma sequences $\{x^{(k)}\}_{1 \leq k \leq M}$ with $N$ tokens each, each sequence is i.i.d. generated by a mean and precision parameter sampled from a prior. For sequence $x^{(k)}$, the generative process is,

$$\tau_k \sim \text{GAMMA}(\alpha_0, \beta_0)$$
$$\mu_k | \tau_k \sim \mathcal{N}(\mu_0, (\lambda_0 \tau_k)^{-1})$$
$$x_n^{(k)} | \mu_k, \tau_k \sim \mathcal{N}(\mu_k, \tau_k^{-1}) \text{ for } n \in \{1, ..., N\},$$

where $\alpha_0, \beta_0, \mu_0, \lambda_0$ are fixed hyperparameters. The full generative processes for the other two models are given in Appendix A.2.

The Gaussian-Gamma model generates from a Gaussian distribution with two unknown parameters, mean $\mu$ and precision $\tau$. The predictive distribution for the next token $x_{n+1}$ is

$$p(x_{n+1} | x_{1:n}) = \int p(x_{n+1} | \mu, \tau) p(\mu, \tau | x_{1:n}) d(\mu, \tau).$$

The optimal Bayesian agent uses the same prior distributions as the Gaussian and Gamma that generate $\mu$ and $\tau$. It will analytically infer the ground truth posterior $p(\mu, \tau | x_{1:n})$ for any stream of data it sees, and use this posterior for predicting next tokens. A Bayesian agent can also use other suitable priors and converge to the optimal posterior. To be consistent with other exchangeable conjugate models, we denote $\theta = (\mu, \tau)$ to indicate latent variables whose posterior distribution is predictive sufficient.

**Discrete hypothesis spaces** Each sequence in this dataset consists of two-dimensional points uniformly sampled from a rectangular region in 2D space (cf. Tenenbaum, 1998). The hypothesis space $\mathcal{H}$ is the set of all rectangles whose corner points are pairs of integers in $\{0, 1, 2, ..., 7\}$. The generative process uniformly samples a rectangle $h_{\text{rect}}$ from the set of all rectangles $\mathcal{H}$. Then, each token in a sequence is sampled uniformly from the region defined by $h_{\text{rect}}$.

#### 4.1.2 Probing experiments

**Implementation details** For each dataset, the out-of-distribution (OOD) hyperparameters are chosen such that OOD data are centered on a different mean and has a different spread. The exact choices are detailed in Appendix A.3, along with hyperparameter sweep in Appendix A.5.

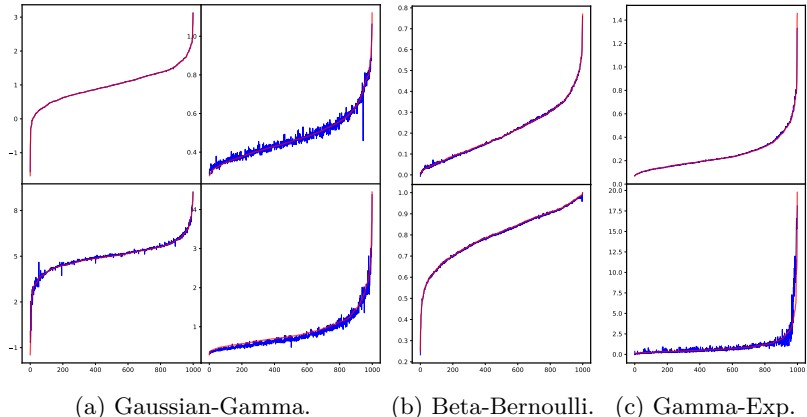

(a) Gaussian-Gamma.          (b) Beta-Bernoulli.   (c) Gamma-Exp.

Figure 2: Probe recovery of transformer-learned sufficient statistic (blue) and ground truth sufficient statistic (red) on the y-axis, across 1000 test datapoints on the x-axis. In the plot above, the datapoints are sorted based on their ground truth sufficient statistic. The first row shows parameters probed in the non-OOD case (from left to right: Gaussian mean $\mu$, Gaussian precision $\tau$, Bernoulli mean, and Exponential mean). The second row shows the corresponding information in the OOD case.

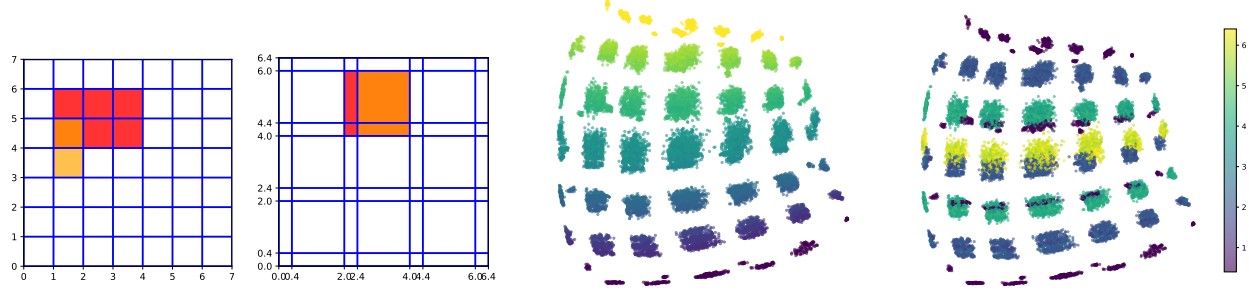

(a) A hypothesis space with equal-width rectangles (left), and unequal widths (right).

(b) Colored by mean along y-axis of the generating rectangle.

(c) Colored by spread along y-axis of the generating rectangle.

Figure 3: (a): Two discrete hypothesis spaces $\mathcal{H}$ used in experiments. Any continuous rectangle contained within the axes (e.g., the red or the orange rectangle) is a valid hypothesis $h \in \mathcal{H}$. The data consist of a sequence of points sampled uniformly from the target rectangle. (b) and (c): Two-dimensional representation of embeddings of all validation datapoints (the setup is unequal width and sample size = 50). The two subfigures show the same embeddings, colored by properties of the true generating rectangles.

In discrete hypothesis spaces, we use rectangles with seven unit blocks on each side, resulting in a size-784 hypothesis space, i.e., 784 possible rectangles from which sequences are drawn. The unit lengths are either 1, or 0.4 and 0.16 alternating (Figure 3a).

**Experimental results**   We hypothesize that a transformer trained on these datasets should come to represent the sufficient statistics of the corresponding distribution.

*Sufficient statistic.* Results for the three Bayesian conjugate models are shown in Figure 2, where the probe successfully decodes the sufficient statistics $\theta$. For hypothesis spaces, we probe the distribution $p(h|x_{1:n})$, which is a length-784 simplex vector. Results are shown in Table 1. In general, the true hypothesis can be found with high accuracy, even though the number of classes is high.

Figure 3 visualizes how embeddings represent each hypothesis $h$ through Principal Components Analysis (PCA) that reduces the 128-dim embeddings to 2D, and the points are colored by properties of each rectangle. Embeddings are clustered into different regions based on the geometry of the true generating rectangle (Figure

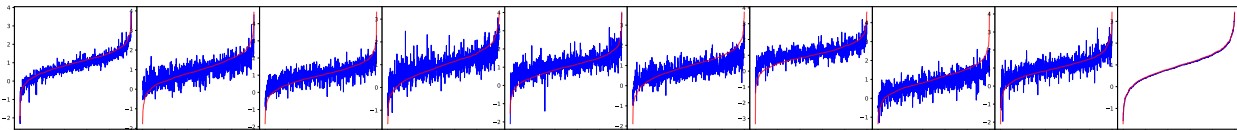

Figure 4: Probing over the first 10 tokens themselves using the 10th token embedding of the transformer. Aside from perfectly encoding the 10th token, this embedding does not show memorization over the other 9 tokens as suggested by the noise in probe recovery.

3b and 3c), where, in 3b for instance, the top yellow clusters correspond to the upper-most rectangles, and dark blue clusters correspond to the lower-most rectangles. Additionally, the embeddings encode the distance between corners of the generating rectangle (Figure 3c). We also find the same color pattern (but rotated 90 degrees) when the horizontal instead of vertical axis is used to color the embeddings.

*Moments of the posterior distribution.* A consequence of knowing the sufficient statistic is finding the true posterior $p(\theta|x_{1:N})$, and we show that the moments of this posterior can be decoded from the transformer embedding (see Figure 8 and 9 in the Appendix). The moments are functions of the mean and variance of the stream of observed data. In these conjugate models, it might be unsurprising for a transformer to encode the mean of the sequence it processes, because the optimal strategy for its loss function is to always predict the mean of the sequence that it sees so far. However, encoding the variance would not be directly related to this strategy and would support the argument that it infers sufficient statistics.

*Out-of-distribution simulations.* The analytical nature of the Bayesian predictor means that it is robust to datasets generated far from the prior – it would simply update its posterior based on the data. Thus, we probe the transformer on out-of-distribution (OOD) datasets that are generated from a distribution of the same form but with distinct hyperparameters. Figure 2, as well as 8a in the Appendix, suggest that the recovery of sufficient statistic and moments of the posterior distribution is generalizable to OOD cases, just as a Bayes-optimal agent would generalize.

**Memorizing the context or storing sufficient statistics of the context?**   A possible confound behind the probe is that, instead of having a probe that uses the transformer-embedded sufficient statistics, the transformer may memorize a set of tokens, and the probe infers quantities of interest from these memorized inputs. We begin by making sure that the sufficient statistics over the first ten tokens can be decoded from the probe on the tenth token in the Gaussian-Gamma case (Figure 11 in Appendix A.5). Then, we probed the token values themselves to look for memorization. Figure 4 suggests that memorization is generally absent. The 10th token embedding recovers the 10th token perfectly, but cannot recover the other 9 tokens. For the other 9 tokens, a correlation exists between probe results and true token values, but some correlation is expected because even a single token can reveal information about the generating distribution. However, the noise suggests that the model is finding sufficient statistics rather than memorizing.

**Embeddings are parsimonious**   Our analysis so far has focused on whether embeddings contain information that allows decoding sufficient statistics. However, our hypothesis is stronger than this: since the sufficient statistics are enough to encode the relevant information from the data, a model need *only* represent that information. To explore whether transformers construct such parsimonious representations, we fixed the models trained on different distributions above and used a multi-layer perceptron (MLP) to directly predict their last-layer, last-token embeddings using the sufficient statistics of the training sequences as input.

Results show that in most cases the sufficient statistics capture well over 50% of variance in the embeddings (Table 6 in Appendix A.5). The results show that sufficient statistics capture a substantial amount of variance in transformer embeddings, establishing that the relationship between embeddings and sufficient statistics runs in both directions: we can decode sufficient statistics from embeddings, and we can predict embeddings from sufficient statistics.

Table 2: Probing target quantities in HMM dataset using different transformer token embeddings. The target $p(z_{n+1}|x_{1:n})$ is a simplex vector found by running the forward-backward algorithm, and the target $\hat{z}_{n+1}$ is a scalar standing for the most likely latent class found by Viterbi algorithm. Average and standard error across 10 random seeds are reported.

| Target Quantity | Embedding | $\delta = 0.5$ | | $\delta = 1$ | |
| --- | --- | --- | --- | --- | --- |
| | | Accuracy ↑ | Squared Loss ↓ | Accuracy ↑ | Squared Loss ↓ |
| $p(z_{n+1}|x_{1:n})$ | $x_{1:n}$ | $90.8 \pm 1.7\%$ | $0.011 \pm 0.004$ | $90.4 \pm 1.9\%$ | $0.011 \pm 0.003$ |
| $p(z_{n+1}|x_{1:n+1})$ | $x_{1:n+1}$ | $86.5 \pm 1.3\%$ | $0.066 \pm 0.011$ | $82.2 \pm 1.8\%$ | $0.067 \pm 0.011$ |
| $p(z_{n+1}|x_{1:n})$ | $x_{1:n+1}$ | $66.6 \pm 6.6\%$ | $0.072 \pm 0.013$ | $65.9 \pm 6.8\%$ | $0.058 \pm 0.011$ |
| $p(z_{n+1}|x_{1:n+1})$ | $x_{1:n}$ | $53.2 \pm 3.5\%$ | $0.356 \pm 0.014$ | $50.8 \pm 3.6\%$ | $0.278 \pm 0.012$ |
| $\hat{z}_{n+1}$ | $x_{1:n}$ | $59.8 \pm 3.8\%$ | / | $61.4 \pm 4.7\%$ | / |
| $\hat{z}_{n+1}$ | $x_{1:n+1}$ | $80.8 \pm 2.2\%$ | / | $77.5 \pm 2.8\%$ | / |

## 4.2 Hidden Markov model

### 4.2.1 Generative model

For an HMM generating data with $M$ sequences with $N$ tokens each, we formulate the generative process as

$$A_c \sim \text{DIRICHLET}_C(\gamma) \text{ for } c \in \{1, ..., C\}$$
$$B_c \sim \text{DIRICHLET}_V(\delta) \text{ for } c \in \{1, ..., C\}$$
$$z_0 \sim \text{CATEGORICAL}(\pi)$$
$$x_i \sim \text{CATEGORICAL}(B_{z_i})$$
$$z_{i+1} \sim \text{CATEGORICAL}(A_{z_i}),$$

where $C, V, \pi$ are initialized and denote, respectively, the number of classes, vocabulary size, and a list of probabilities on the number of classes to initialize the first latent state. $\gamma, \delta$ are scalar hyperparameters that are also initialized and fixed, and they represent the evenness of the samples from the Dirichlet distributions. $A$, $B$, and $z$, as a result, represent the transition matrix, the emission matrix, and the latent states, respectively.

We use the forward-backward algorithm (Rabiner, 1989) to compute the posterior $p(z_{n+1}|x_{1:n})$, and explore whether this distribution can be decoded from the transformer embedding on $x_{1:n}$.

### 4.2.2 Probing experiments

**Implementation details**  We choose $C = 4, V = 64, \gamma = 0.5$, and set $\pi$ to be uniform in our experiments. We also vary $\delta$ to control the level of difficulty: how distinct is one class from another.

**Results**  Our theoretical treatment suggests that the transformer should encode the predictive sufficient statistic $p(z_{n+1}|x_{1:n})$ or $p(z_n|x_{1:n})$ . The latter is the target used in belief state inference with transformers (Shai et al., 2024). However, there are other natural decoding targets that could be used. A simple target is the one-hot vector corresponding to the most likely hidden state $\hat{z}_{n+1}$. This is not a predictive sufficient statistic and does not encode the full information about the sequence relevant to future prediction. Our analysis thus suggests that it will thus provide a poorer match to the information contained in the embedding.

Table 2 suggests that the transformer encodes the predictive sufficient statistics. Furthermore, performance is better for our hypothesized measure (first and second row) than for related quantities (all other rows). Additionally, it is more difficult to recover the measure used in belief state inference (second row), although it is intuitively easier to learn: the target probability is over the same latent variable, but the emission of this latent variable is observed, unlike in our hypothesized measure where it is unobserved. This may be because if the embedding were to directly encode $p(z_n|x_{1:n})$, the transformer output layer after the embedding would additionally need to encode the transition matrix and additional integration computations, potentially making the task more challenging than encoding $p(z_{n+1}|x_{1:n})$.

Table 3: Topic prediction by the autoregressive transformer (AT), BERT, LDA, and word-embedder (WE) on LDA-generated synthetic datasets, along with standard deviations across 3 random seeds. Hyperparameter $\alpha$ defines the dataset generation process, where a higher $\alpha$ means a more difficult task with underlying topics being more evenly distributed. AT and BERT have similar performance in the easiest setting, but AT performs well in harder settings where BERT performances worsen. End-to-end WE achieves stronger performance than language models, and LDA matches expectations by providing an upper bound in performance.

| $\alpha$ | Method | Accuracy ↑ | L2 loss ↓ | Tot. var. loss ↓ |
|---|---|---|---|---|
| 0.5 | AT | $82.8\% \pm 0.5\%$ | $0.041 \pm 0.001$ | $0.141 \pm 0.001$ |
| | BERT | $83.6\% \pm 1\%$ | $0.036 \pm 0.003$ | $0.131 \pm 0.005$ |
| | LDA | $87\% \pm 0.6\%$ | $0.029 \pm 0$ | $0.117 \pm 0.001$ |
| | WE | $85.8\% \pm 1.3\%$ | $0.03 \pm 0.001$ | $0.119 \pm 0.002$ |
| 0.8 | AT | $75.5\% \pm 0.8\%$ | $0.044 \pm 0.001$ | $0.144 \pm 0.001$ |
| | BERT | $51.5\% \pm 1.7\%$ | $0.111 \pm 0.005$ | $0.233 \pm 0.011$ |
| | LDA | $82.6\% \pm 0.5\%$ | $0.036 \pm 0.001$ | $0.133 \pm 0.004$ |
| | WE | $80.9\% \pm 0.5\%$ | $0.029 \pm 0$ | $0.116 \pm 0.001$ |
| 1 | AT | $70.5\% \pm 1.6\%$ | $0.045 \pm 0.001$ | $0.146 \pm 0.003$ |
| | BERT | $46.6\% \pm 3.3\%$ | $0.1 \pm 0.004$ | $0.222 \pm 0.006$ |
| | LDA | $79.6\% \pm 1.4\%$ | $0.045 \pm 0.004$ | $0.147 \pm 0.006$ |
| | WE | $79.4\% \pm 1\%$ | $0.027 \pm 0$ | $0.113 \pm 0.001$ |

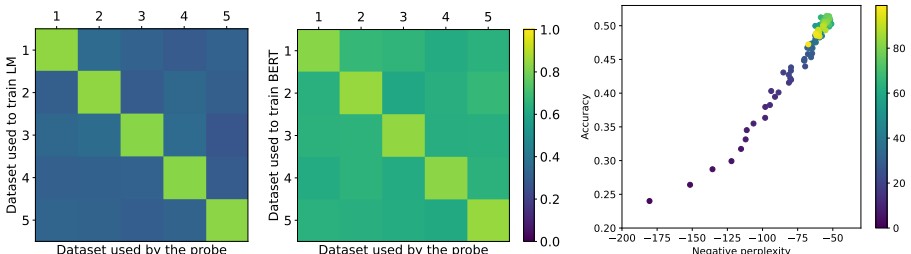

(a) AT probe accuracies.  (b) BERT probe accuracies.  (c) Accuracy vs. neg. perplexity

Figure 5: Figure 5a and 5b: control experiments showing AT (left) and BERT's (middle) probe validation performance on synthetic data. For each AT and BERT, five models are trained and validated on five datasets, with each dataset generated by a distinct topic model. Colors show probe accuracy. A cell on row $i$ and column $j$ corresponds to model $i$ on dataset $j$, so the diagonal corresponds to a model on its own dataset. *For AT, performance is only strong on the dataset with the same generating topic model, suggesting that the underlying statistical model, not the probe taking different word embeddings, is responsible for performance – a relationship that is also present for BERT but to a lesser degree.* Figure 5c: 20NG probe classification performance (accuracy) vs. negative perplexity measured at 100 different tokens. The dots are colored by the position percentile. *Probe performance increases with lower perplexity.*

## 4.3  Topic Models

We aim at recovering from language models the topic mixture $\theta_i$ that draws words in document $i$.

### 4.3.1  Probing experiments

**Experiment setup**  The dataset is bags-of-words generated by LDA. We set the vocabulary size $V = 10^3$, number of topics $K = 5$, and generated $N = 10^4$ documents that are each 100 words long.

**Models**  We trained four models: an autoregressive transformer decoder (AT), BERT, LDA, and an end-to-end word embedder (WE). BERT uses a masked objective instead of the autoregressive one, and is implemented by a small version called BERT-TINY (Turc et al., 2019). LDA is used to establish an upper-bound for model performance. We also include a model intended to provide an upper bound for embedding performance:

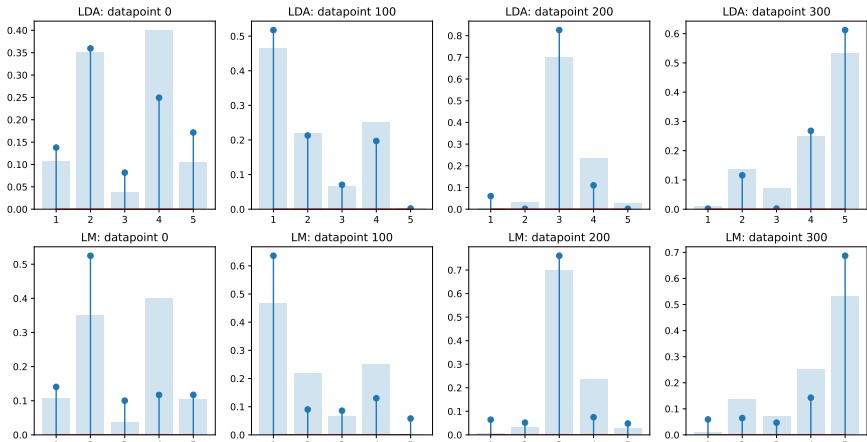

Figure 6: Distribution of synthetic data topics predicted by LDA (row 1) and AT classifier (row 2) for different validation datapoints. Predictions are stick plots, and ground truth is bar chart in the background. They both exhibit learning of topic mixtures by trying to match the distribution, in addition to top-1 agreement.

a word embedder which is a matrix that maps from the vocabulary space to the AT / BERT embedding dimension and is end-to-end trained with the probe that predicts topics.

For each model, the hidden sizes and final-layer embedding sizes are 128. AT has 4 decoder layers to match the size of BERT-TINY. More hyperparameters are detailed in Appendix A.3.

**Metrics**   We use accuracy, L2 loss, and total variation loss to measure both classification performance and recovery of the topic mixture distribution. Accuracy is defined as how often the top topic predicted by the classifier's mixture agrees with the top topic from ground truth. The remaining loss measures apply to the whole topic mixture.

**Results**   Figure 6 shows five examples of ground truth, transformer-predicted, and LDA-predicted topic mixtures. Table 3 shows that all models demonstrate success at recovering latent topics on at least the easiest setting (i.e., $\alpha = 0.5$), being able to return both top-topic accuracy and the topic distribution spread. Between AT and BERT, the probe on AT is able to infer latent topic structures in more difficult tasks (i.e., $\alpha = 0.8, 1$) whereas the probe on BERT shows deteriorating performance. LDA outperforms both AT and BERT as expected because it is specified exactly to learn a dataset generated by the other manually initialized LDA. However, the strong WE performance suggests that the probe can predict topics mainly from stand-alone words. This raises the question whether AT and BERT are learning an underlying statistical model, or are simply uniquely embedding each word and making the probe mainly responsible for topic recovery.

### 4.3.2   Controlling for probe performance

In this section, we conduct control experiments that suggest that language models learn an underlying statistical model, making them—rather than the topic probes taking word embeddings —mainly responsible for successful topic recovery. If AT or BERT performs well just because it gives each word a unique embedding from which a trained probe suffices to recover the topic mixture, then a probe on top of the language model should additionally predict topic mixtures from a different underlying topic model than the one that generated the language model's training data.

To assess this, we generate five datasets using five distinct topic models under the setting of $\alpha = 0.5$. By definition, these topic models initialized five different sets of topics, or distributions over words $\{\beta_1, ..., \beta_K\}_j, j \in \{1, 2, 3, 4, 5\}$. The transformer trained on set $j$ should encode the mixture of topics $\theta$ that corresponds to topics $\{\beta_1, ..., \beta_K\}_j$.

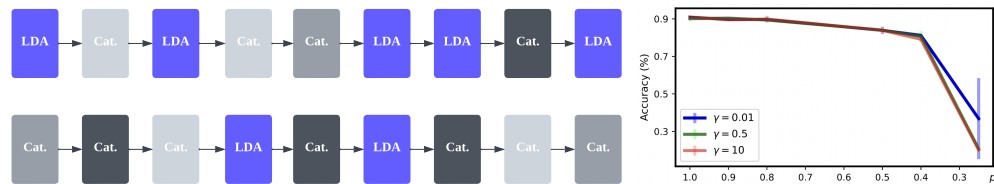

Figure 7: Left: Illustration of two sequences (upper: $p \approx 0.5$; lower: $p \approx 0.25$). Each box is a word, coming from either the LDA-generated semantic component (blue), or one of the three categorical distributions that indicate three syntactic classes (grey). Right: Probing accuracy across semantic proportion $p$ on HMM-LDA generated datasets. Lower $\gamma$ corresponds to more sparse transition matrices. *The language model successfully encodes semantics if it occupies a considerable enough proportion.*

Table 4: 20NG topic prediction performance based on different LLMs. *Trained LLMs substantially outperform the null GPT-2 model, supporting the hypothesis that the training process encourages LLMs to implicitly develop topic models. Autoregressive models (the first five models) statistically significantly outperform non-autoregressive models (the next two).*

| | | $K = 20$ | | | $K = 100$ | | |
|---|---|---|---|---|---|---|---|
| Model | Parameters | Accuracy ↑ | L2 loss ↓ | Tot. var. loss ↓ | Accuracy ↑ | L2 loss ↓ | Tot. var. loss ↓ |
| GPT-2 | 124M | $61.4\% \pm 1.5\%$ | $0.106 \pm 0.002$ | $0.211 \pm 0.001$ | $42.3\% \pm 2.4\%$ | $0.097 \pm 0$ | $0.192 \pm 0.001$ |
| GPT-2-MEDIUM | 355M | $62.6\% \pm 1.7\%$ | $0.104 \pm 0.002$ | $0.209 \pm 0.002$ | $42.9\% \pm 2.4\%$ | $0.096 \pm 0$ | $0.19 \pm 0.001$ |
| GPT-2-LARGE | 774M | $62.4\% \pm 1.8\%$ | $0.102 \pm 0.002$ | $0.208 \pm 0.002$ | $43.1\% \pm 2.3\%$ | $0.095 \pm 0.001$ | $0.189 \pm 0$ |
| LLAMA 2 | 7B | $62.6\% \pm 1.7\%$ | $0.101 \pm 0.002$ | $0.206 \pm 0.002$ | $43.3\% \pm 2.4\%$ | $0.095 \pm 0.001$ | $0.189 \pm 0.001$ |
| LLAMA 2-CHAT | 7B | $62.9\% \pm 1.7\%$ | $0.102 \pm 0.002$ | $0.207 \pm 0.002$ | $43.2\% \pm 2.5\%$ | $0.095 \pm 0.001$ | $0.189 \pm 0$ |
| BERT | 110M | $56.3\% \pm 1.5\%$ | $0.113 \pm 0.003$ | $0.222 \pm 0.003$ | $38.6\% \pm 2.5\%$ | $0.1 \pm 0.001$ | $0.191 \pm 0.001$ |
| BERT-LARGE | 336M | $55.2\% \pm 1.2\%$ | $0.116 \pm 0.002$ | $0.226 \pm 0.003$ | $38.9\% \pm 2.9\%$ | $0.1 \pm 0.001$ | $0.191 \pm 0.001$ |
| Null GPT-2 | 124M | $27.3\% \pm 1\%$ | $0.209 \pm 0.003$ | $0.322 \pm 0.005$ | $13.8\% \pm 1.7\%$ | $0.145 \pm 0.001$ | $0.248 \pm 0.003$ |

One AT and one BERT is trained on each dataset. On each model, five probes are used to predict topics from each of the five datasets. Results are shown in Figure 5. AT shows a strong distinction between predicting its own dataset versus datasets from other topic models, whereas this distinction is present but weaker for BERT. These results are evidence that AT and BERT indeed encode topic information because, if the topic information were instead constructed by the probe, the probe would work equally well on mismatched datasets as on matched ones.

## 4.4 From HMM-LDA to Natural Corpora

In this section, we start with a synthetic dataset with a more realistic assumption than topic models — that topics are an exchangeable component in a partially exchangeable sequence. We generate a dataset based on both LDA and a hidden Markov model (HMM). Then, we move to evaluate the extent to which LLMs recover topic mixture two natural corpora, *20Newsgroups* (20NG) and *WikiText-103*.

### 4.4.1 Topic distribution can be recovered when LDA is a sequence sub-component

Most natural texts are not exchangeable. For example, consider this sentence: "The **sediment** found in the **quartz** includes **silicon**." This sentence is neither fully exchangeable nor generated purely by a latent state model. However, if we only consider the words that contribute to the sentence's topic (geology), we are left with **sediment**, **quartz**, **silicon**. These words can be plausibly generated by an exchangeable topic model as a sub-component: the sentence "The **silicon** found in the **sediment** includes **quartz**" also has the topic of geology even though word order changes. While we do not exhaustively list possible factors behind language that can be embedded by LLMs, we study to what extent can topics be encoded by model embeddings under settings where topical words occupy a varied proportion with respect to the whole sequence.

**Model**  Similar to the LDA experiments, we generate synthetic documents with an HMM-LDA model (Griffiths et al., 2004) (Figure 7). HMM-LDA combines semantics and syntax, and posits that each word comes from a latent class and that class transitions are governed by an HMM. Using the HMM-LDA model allows one to manipulate the degree to which topic exists as a sub-component of the sequence.

When $p$ is 1, all the words are generated from the LDA model. As $p$ decreases, more words are generated from the syntactic classes. We can then examine how well topic distributions are recovered from the embeddings of models that are trained on data generated from HMM-LDA models that vary $p$.

**Results**  Results in Figure 7 suggest that without exchangeability, the autoregressive transformer still learns the same semantic latent variables provided there is a large enough proportion of LDA-generated component (i.e., $p \geq 0.4$). This suggests that the theory of finding explainable quantities in autoregressive models can potentially be extended to real texts where interpretable probabilistic models, such as the topic model, exist as a sub-component of language.

### 4.4.2  Recovering latent topic distributions in natural corpora

This section turns to analyzing LLMs pretrained on natural language. Although an exhaustive list of predictive sufficient statistic is difficult to identify, topical words do form a proportion of factors underlying the text. This allows us to hypothesize that topic mixtures can be decoded from the transformer residual streams.

**Datasets**  We use *20Newsgroups* (20NG) and WikiText-103 (Merity et al., 2016). 20NG is a collection of 18,000 posts written in a style similar to informal emails, divided into 20 subjects. Contrasting with the informal language style in 20NG, WikiText-103 consists of over 100 million tokens sourced from the set of verified articles on Wikipedia that are classified as Good and Featured.

**Setup**  We train LDA models across three random seeds on each dataset, and use pretrained large language models as our LLM. Specifically, the LLMs are GPT-2, GPT-2-MEDIUM, and GPT-2-LARGE (Radford et al., 2019); LLAMA 2 and LLAMA 2-CHAT (Touvron et al., 2023); and BERT, and BERT-LARGE (Devlin et al., 2019). The probes target LDA-learned topic mixtures with $K \in \{20, 100\}$. For the models, we additionally include a randomly initialized GPT-2, called Null GPT-2, as a control that can differentiate the contribution of the LLM from the contribution of the probe. Considering that the first token may lead to a useful embedding that stands for the ⟨CLS⟩ token in BERT series, we searched across the first, last, and average embeddings.

**Topic prediction results**  Performance is shown in Table 4 for 20NG and in Table 7 in the Appendix for Wikitext-103. To predict mixtures of twenty topics, random guessing would yield a 5% accuracy for $K = 20$ or 1% accuracy for $K = 100$. The best-performing LLM on each dataset demonstrates success at encoding topic distributions by achieving 88.5%/74.2% accuracy on WikiText-103, and 62.9%/43.2% on 20NG. The probes here take averaged token embeddings on the last layer in each LLM. While having direct information over whole sequences is advantageous compared to over an arbitrary token, we show that using only the last token preserves strong performance, with 73.7%/58.9% accuracy on WikiText-103, and 52.2%/34.7% on 20NG (Table 14 and 13 in Appendix). These results suggest that topic mixtures are directly encoded in embeddings, rather than resulting from high-quality word representations.

**Topics encoded by inner layers**  We also explored the possibility of decoding topic mixtures from other LLM layers, as opposed to only the last layer (Table 8 in Appendix A.5). In the LLAMA-2 series, the decodability of topic mixtures progressively increases as we move from the word embedder to the intermediate layers. Although a word embedder can potentially achieve high accuracy as demonstrated by the synthetic dataset, we observe that LLAMA-2 relies on its inner layers to encode topic mixtures. This phenomenon can be caused by LLAMA-2's stronger ability to model the next word, which is correlated with the ability to capture latent structures, which we discuss in more detail in the next paragraph.

**Probe performance and LLM perplexity**  To show that the the posterior predictive on topics corresponds to autoregressive prediction, we analyze 100 probes trained on 100 different token positions, along with their corresponding LLM perplexity (Figure 5c). Each position is defined based on the corresponding percentile

of the total document (each document has a different length). We expect that as perplexity on a token increases, probe performance based on the embedding taken from that token would decrease. This hypothesis is supported by the linear trend in Figure 5c. A possible confound in this result is that perplexity tends to decrease as the position in a document increases, so the results could be driven by position rather than perplexity; however, we ran a linear mixed-effects model and found that perplexity continued to have a statistically significant effect controlling for position (see Appendix A.4).

## 5 Discussion

In this paper, we have shown that autoregressive modeling is connected to Bayes optimal agents, where the sufficient statistic is encoded by the model in two major cases: 1) exchangeable distributions that emcompass a wide range of generative models including the topic model; and 2) latent state models that include the HMM. We have also seen that when these latent variables form a significant subcomponent of the data, they can still be encoded, leading to an explanatory approach on why topic mixtures are embedded in transformer residual streams.

There are several potential implications and future directions of our findings.

**Interpretability.** Understanding the inner workings of LLMs is important for AI safety and trustworthiness. However, the sheer size of LLMs makes it challenging to analyze. Our results suggest that interpretable models of document structure - such as topic models - provide useful guidance about what mechanistic interpretability should look for. This in turn means that one goal for interpretability work should be enhancing models like LDA because such advances will in turn sharpen our ability to interpret LLM representations.

**Reverse direction - training LLMs to capture inductive biases.** We have shown how autoregressive models capture latent structures when predicting the next token. Another direction to explore is whether it is possible to explicitly incorporate inductive biases during training to more reliably and efficiently capture these latent structures, as have been done in modeling language (McCoy & Griffiths, 2025) and vision (Carballo-Castro et al., 2024).

**Extensions to masked language models.** It has been shown that the autoregressive objective can be broken down into the Bayesian predictive distribution in several cases. It is possible to extend this breakdown to masked language models as well (Appendix A.1). Whereas the autoregressive model learns a different posterior distribution $p(\theta|x_{1:n})$ for each $n$ as it progresses through tokens in a sequence, the masked model learns that same $p(\theta|x_U)$, where $U$ denotes the set of unmasked indices, resulting in a less expressive objective. On synthetic topic model bags-of-words, the masked model underperforms, but it is worth exploring whether its performance is consistently worse than that of the autoregressive model in a wider range of cases.

**Applications.** Our results help inform construction and use of embeddings in practice, and can be applied to generative models that satisfy the conditions discussed by the paper. The analyses can be naturally extended to other latent variables that do not depend heavily on word order, such as the author type of the document (Andreas, 2022) or the author's sentiment (Radford et al., 2017).

As another example, our results can also inform time series modelling. A practitioner might have a time series dataset with underlying factors that are informed by human experts. If the practitioner wants to use deep autoregressive models to construct an embedding that contains a certain kind of information from the input, they may need to ensure that this information is a predictive sufficient statistic for the task.

## 6 Conclusion

We have developed a general framework for analyzing what the embeddings of an autoregressive language model should represent. Our analyses suggest that such embeddings should represent latent structures such that the next token $x_{n+1}$ is independent from previous tokens $x_{1:n}$ when conditioned on that structure, a property possessed by predictive sufficient statistics. We confirmed this hypothesis with probing experiments

on cases where predictive sufficient statistics can be identified. We hope that our findings contribute to bridging the gap between Bayesian probabilistic models and deep neural networks.

**Acknowledgments**

This work was supported by ONR grant number N00014-23-1-2510. TRS is supported by an NDSEG Fellowship. RTM was supported by the NSF SPRF program under Grant No. 2204152.

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

# A Appendix

## A.1 Masked language models

Masked language models (MLMs) are another class of LLMs that have been successful at modeling text, and we explore whether the objective of MLMs is also equivalent to implicit Bayesian inference under the exchangeability assumption. We find that it is, but it differs from autoregressive language models in a way that results in less expressivity.

MLMs, notably including the BERT series (Devlin et al., 2019), randomly mask certain tokens in the input sequence, and the model's goal is to predict the original words at the masked indices only by considering their surrounding context. Unlike autoregressive models, the MLM does not model a coherent joint distribution of the data (Yamakoshi et al., 2022; Young & You, 2022). However, the log objective can be extended as follows,

$$\mathcal{L}_{MLM}(x_{1:N}) = \sum_{n \in M} \log p(x_n | x_{i, i \in U})$$
$$= \sum_{n \in M} \log \int p(x_n | \theta) p(\theta | x_{i, i \in U}) d\theta. \tag{4}$$

where $M$ denotes the set of masked indices, and $U$ denotes the set of unmasked indices. The proof is given below, and is a simple extension of the derivation for the autoregressive version. The difference between the MLM objective and the autoregressive objective is that in the summation, the prediction of each token $x_n$ uses the same posterior over the latent variable $p(\theta | x_{i, i \in U})$. In other words, each token $x_n$ is predicted independently from the latent variable $\theta$. As a result, MLM forms a less expressive Bayesian inference objective than autoregressive models. Therefore, we aim to empirically evaluate both the ability of MLM to recover latent variables, and whether its performance differs from that of autoregressive models.

We now prove the equivalence between the MLM objective and Bayesian inference in Equation 4. The statement is that, given an exchangeable process $x_1, x_2, ..., x_n$,

$$\mathcal{L}_{MLM}(x_{1:N}) := \sum_{n \in M} \log p(x_n | x_{i, i \in U}) \tag{5}$$

$$= \sum_{n \in M} \log \int p(x_n | \theta) p(\theta | x_{i, i \in U}) d\theta, \tag{6}$$

where $M$ is the set of masked indices and $U$ is the set of unmasked indices, and by construction $M \cap U = \emptyset$.

**Proof.** We first prove the autoregressive version of the equivalence, which Korshunova et al. (2018) proposes for each individual term in the summation but briefly mentions why it is equivalent,

$$\log p(x_{1:N}) = \log p(x_1) + \sum_{n=1}^{N-1} \log p(x_{n+1} | x_{1:n})$$
$$= \log p(x_1) + \sum_{n=1}^{N-1} \log \int_\theta p(x_{n+1} | \theta) p(\theta | x_{1:n}) d\theta. \tag{7}$$

To do so, we shall prove the equivalence in each term in the summation, that is, the statement that

$$p(x_{n+1} | x_{1:n}) = \int_\theta p(x_{n+1} | \theta) p(\theta | x_{1:n}) d\theta \tag{8}$$

for each $n > 1$. First, de Finetti's theorem states that, under the same exchangeability condition,

$$p(x_{1:n+1}) = \int_\theta p(\theta) \prod_{i=1}^{n+1} p(x_i | \theta) d\theta, \tag{9}$$

and also that each $x_i$ is conditionally independent given $\theta$ for all $i$. To show the equivalence, we first divide each side of Equation 9 by $p(x_{1:n})$, assuming that $p(x_{1:n}) > 0$. The left hand side becomes,

$$\frac{p(x_{1:n+1})}{p(x_{1:n})} = p(x_{n+1}|x_{1:n}). \tag{10}$$

The right hand side becomes,

$$\int_\theta p(\theta) \frac{\prod_{i=1}^{n+1} p(x_i|\theta)}{p(x_{1:n})} d\theta = \int_\theta p(\theta) \frac{p(x_{n+1}|\theta) p(x_{1:n}|\theta)}{p(x_{1:n})} d\theta \tag{11}$$

$$= \int_\theta p(x_{n+1}|\theta) \frac{p(\theta) p(x_{1:n}|\theta)}{p(x_{1:n})} d\theta \tag{12}$$

$$= \int_\theta p(x_{n+1}|\theta) p(\theta|x_{1:n}) d\theta. \tag{13}$$

Line 11 uses conditional independence to combine the product on $i = 1$ through $n$, and line 13 uses Bayes rule. Therefore, because of Equation 9, we prove the statement of Equation 8.

The proof for MLM in Equation 6 can be shown by a simple extension. We shall use $x_U$ as short hand for $x_{i,i \in U}$. Using de Finetti's theorem,

$$p(x_{\{n\} \cup U}) = \int_\theta p(\theta) \prod_{i \in \{n\} \cup U} p(x_i|\theta) d\theta, \tag{14}$$

Dividing each side of the above equation by $p(x_U)$, the left hand side becomes,

$$\frac{p(x_{\{n\} \cup U})}{p(x_U)} = p(x_n|x_U). \tag{15}$$

The right hand side becomes,

$$\int_\theta p(\theta) \frac{\prod_{i \in \{n\} \cup U} p(x_i|\theta)}{p(x_U)} d\theta = \int_\theta p(\theta) \frac{p(x_n|\theta) p(x_U|\theta)}{p(x_U)} d\theta \tag{16}$$

$$= \int_\theta p(x_n|\theta) \frac{p(\theta) p(x_U|\theta)}{p(x_U)} d\theta \tag{17}$$

$$= \int_\theta p(x_n|\theta) p(\theta|x_U) d\theta. \tag{18}$$

Same as in the proof for the autoregressive version, line 16 uses conditional independence to combine the product on $U$, and line 18 uses Bayes rule. Therefore, we have that $p(x_n|x_{i,i \in U}) = \int p(x_n|\theta) p(\theta|x_{i,i \in U}) d\theta$. Thus, we have shown the equivalence for each term in the summation of the original statement Equation 6.

## A.2 Definition of additional exchangeable conjugate models

We consider the generative process for sequence $x_i$, where $x_{ij}$ are i.i.d. across $j$.

### A.2.1 Topic model

The generative model for LDA is

For each topic $k$ in $(1, ..., K)$,

    1. Draw topic $\beta_k \sim \text{Dirichlet}_V(\eta)$.

For each document $i$,

    1. Draw topic mixture $\theta_i \sim \text{Dirichlet}(\alpha)$.

    2. For each word $j$ in document $i$,

        (a) Draw topic assignment $t_{ij} \sim \text{Categorical}(\theta_i)$,

        (b) Draw word $x_{ij} \sim \text{Categorical}(\beta_{t_{ij}})$,

where $V$ is the vocabulary size, and $\alpha$ and $\eta$ are pre-initialized hyperparameters.

### A.2.2 Beta-Bernoulli model

$$\theta_i \sim \text{BETA}(\alpha, \beta)$$
$$x_{ij} \sim \text{BERNOULLI}(\theta_i),$$

where $\alpha, \beta$ are fixed hyperparameters.

### A.2.3 Gamma-Exponential model

$$\theta_i \sim \text{GAMMA}(\alpha, \beta)$$
$$x_{ij} \sim \text{EXPONENTIAL}(\theta_i),$$

where $\alpha, \beta$ are fixed hyperparameters.

### A.2.4 HMM-LDA

There are $C$ classes $\{c_1, ..., c_C\}$ which follow an HMM transition matrix. One class is the semantic class $c_1$, and a word coming from $c_1$ is generated by an LDA model. Words of other classes are 'syntactic' and are directly generated by a distribution over words with a Dirichlet prior, one distribution for each $\{c_2, ..., c_C\}$. The probability of generating from $c_1$, the LDA class, is determined by a parameter $p$ which specifies the transition probability into that class from all other classes. The remaining transition probabilities are generated from a Dirichlet($\gamma$) prior; lower $\gamma$ corresponds to more sparse transition matrices.

### A.3 Implementational details

All computations for synthetic datasets are run on single Tesla T4 GPUs, and those for natural corpora are run on single A100 GPUs.

**Experimental process**   Each dataset is split into three sets: set 1, set 2, and set 3. Set 1 is used for training the transformer. Set 2 is used for validating the transformer and getting embeddings from transformer that are used to train the probe. Set 3 is used for validating the probe.

Except discrete hypothesis space datasets and natural corpora, the sizes for the three sets are: 10000, 3000, 1000, and each sequence is 500-tokens long. In the discrete hypothesis space datasets, we experimented with different sequence lengths (detailed in our results), and the sizes for the three sets are: 20000, 19000, 1000.

In HMM-LDA, sequence lengths are 400, and the sizes for the three sets are 10000, 1000, 1000.

On 20NG, probe training and validation are run on 11,314 and 7,532 documents, respectively. On WikiText-103, probe training and validation are run on 28,475 and 60 documents, respectively. Both splits are derived directly from train-validation split provided by the dataset sources. Note that set 1 is not used for natural corpora because we use pretrained LLMs.

**Transformer**   Except for topic models, we use a three-layer transformer decoder with hidden-size $= 128$ and number of attention heads $= 8$. If the input is categorical (similar to tokens in natural corpus), we employ the standard word embedder layer before the decoder layers. If the input is continuous, we use a Linear layer to map inputs to dimension 128 in place of the word embedder layer.

Dropout $= 0.1$ is applied, and learning rate $= 0.001$, batch-size $= 64$.

**Transformers on topic models**   Autoregressive transformer (AT) and BERT hyperparameters for training are given in Table 9. Configurations of BERT are identical to those of BERT-TINY from Turc et al. (2019). AT hidden sizes and final layer embedding sizes are 128, same as BERT-TINY, and it uses four layers, resulting in 655,336 parameters. Bert has 608,747 parameters.

**Probe**  The probe is a linear layer with softmax activations. Learning rate is tuned in $[0.001, 0.01]$, and batch-size $= 64$.

**Hyperparameters for exchangeable conjugate models**  The Gaussian-Gamma hyperparameters $\alpha_0, \beta_0, \mu_0, \lambda_0$ are $\{5, 1, 1, 1\}$. The OOD hyperparameters are $\{2, 1, 5, 1\}$.

On the Beta-Bernoulli model, we use $\alpha = 2, \beta = 8$. In the OOD case, $\alpha = 8, \beta = 2$.

On the Gamma-Exponential model, we use $\alpha = 2, \beta = 4$. In the OOD case, $\alpha = 2, \beta = 1$.

## A.4   Linear mixed-effects model

We want to validate our hypothesis that the LLM's latent topic representation helps it predict individual tokens. While Fig. 5c is suggestive of this relationship, here we use statistical testing to confirm it.

Concretely, we use a linear mixed-effects model to predict the per-token perplexity. We analyze 701,243 individual tokens from 20NG test corpus using GPT-2. Perplexity naturally decreases as the LLM processes the document, so we include a fixed effect of token position and a random effect for the document itself; finally, we include the topic decoding accuracy (a binary 0 / 1 outcome based on the topic probe) as the variable of interest. We extract 100 tokens per document, stratified so they are evenly spaced, and represent the token position as the percent into the document,

$$\text{perplexity} \sim \text{token\_position} + \text{topic\_accuracy} + (1|\text{document\_id}). \tag{19}$$

We find significant effects for both token position and topic accuracy,

| Effect | Group | Term | Estimate | Std. Error | Statistic | DF | p-value |
|---|---|---|---|---|---|---|---|
| fixed | | (Intercept) | 4.65 | 0.01 | 413.28 | 21078.63 | <2e-16 |
| fixed | | topic_accuracy | -0.15 | 0.01 | -16.51 | 355354.64 | <2e-16 |
| fixed | | token_position | -0.78 | 0.01 | -58.47 | 696289.69 | <2e-16 |
| ran_pars | document_id | sd__(Intercept) | 0.63 | | | | |
| ran_pars | Residual | sd__Observation | 3.22 | | | | |

Finally, we obtain a Variance Inflation Factor of 1.014742 between accuracy and token position, suggesting an acceptable degree of colinearity between the two variables.

## A.5   Additional results

**Hyperparameter sweep**  Table 5c shows hyperparameter sweep in the Bayesian conjugate models setting, across transformer embedding size in $\{8, 32, 128\}$, number of layers in $\{2, 3, 4\}$, and number of attention heads in $\{4, 8\}$. In general, we observe that embedding size affects performance most significantly. In the Gaussian-Gamma and Bernoulli datasets, performance improves with higher embedding size. In the Gamma-Exponential dataset, performance is best with embedding size $= 32$.

Hyperparameters for probes on AT and BERT on LDA are given in Table 10.

Hyperparameters for probes on the LLMs in natural corpora are given in Table 11 and Table 12.

**Moments**  Figure 8a shows that the probe decodes the moments of the posterior distribution of the Gaussian-Gamma model. Because the higher distribution moments are more volatile in value and less directly related to estimating the parameters $\theta$, we examine whether existing discrepancies are caused by an overly simple probe. We perform a second set of experiments where the probe has a hidden layer with ReLU activations (Figure 8b), showing stronger alignment.

Figure 9 shows results on posterior distribution moments on Beta-Bernoulli and Gamma-Exponential models. On the Gamma-Exponential model, we divide the target second, third, and fourth moments by factors of 10, 100, 1000, respectively so that each moment is given roughly equal importance.

Table 5: Scaled MSE across hyperparameter settings where the probe targets sufficient statistics, along with standard error across three random seeds. In the Gaussian-Gamma case, we report only on the second sufficient statistic, i.e., the standard deviation of the seen sequence (which is more challenging than the mean), to avoid clutter.

(a) Gaussian-Gamma.

| | num heads = 4 | | | num heads = 8 | | |
|---|---|---|---|---|---|---|
| | $d = 8$ | $d = 32$ | $d = 128$ | $d = 8$ | $d = 32$ | $d = 128$ |
| 2 layers | $0.168 \pm 0.022$ | $0.076 \pm 0.023$ | $0.069 \pm 0.013$ | $0.303 \pm 0.039$ | $0.045 \pm 0.010$ | $0.050 \pm 0.005$ |
| 3 layers | $0.566 \pm 0.017$ | $0.066 \pm 0.018$ | $0.048 \pm 0.005$ | $0.321 \pm 0.014$ | $0.034 \pm 0.005$ | $0.046 \pm 0.010$ |
| 4 layers | $0.451 \pm 0.059$ | $0.056 \pm 0.008$ | $0.047 \pm 0.009$ | $0.490 \pm 0.046$ | $0.081 \pm 0.006$ | $0.033 \pm 0.004$ |

(b) Beta-Bernoulli.

| | num heads = 4 | | | num heads = 8 | | |
|---|---|---|---|---|---|---|
| | $d = 8$ | $d = 32$ | $d = 128$ | $d = 8$ | $d = 32$ | $d = 128$ |
| 2 layers | $0.010 \pm 0.002$ | $0.002 \pm 0.000$ | $0.001 \pm 0.000$ | $0.007 \pm 0.002$ | $0.002 \pm 0.001$ | $0.001 \pm 0.000$ |
| 3 layers | $0.008 \pm 0.002$ | $0.001 \pm 0.000$ | $0.000 \pm 0.000$ | $0.010 \pm 0.001$ | $0.001 \pm 0.000$ | $0.001 \pm 0.000$ |
| 4 layers | $0.007 \pm 0.001$ | $0.001 \pm 0.000$ | $0.000 \pm 0.000$ | $0.010 \pm 0.002$ | $0.001 \pm 0.000$ | $0.000 \pm 0.000$ |

(c) Gamma-Exponential.

| | num heads = 4 | | | num heads = 8 | | |
|---|---|---|---|---|---|---|
| | $d = 8$ | $d = 32$ | $d = 128$ | $d = 8$ | $d = 32$ | $d = 128$ |
| 2 layers | $0.004 \pm 0.001$ | $0.001 \pm 0.000$ | $0.009 \pm 0.006$ | $0.002 \pm 0.001$ | $0.001 \pm 0.001$ | $0.067 \pm 0.006$ |
| 3 layers | $0.003 \pm 0.001$ | $0.001 \pm 0.000$ | $0.006 \pm 0.004$ | $0.002 \pm 0.000$ | $0.001 \pm 0.000$ | $0.058 \pm 0.006$ |
| 4 layers | $0.002 \pm 0.000$ | $0.000 \pm 0.000$ | $0.021 \pm 0.008$ | $0.002 \pm 0.000$ | $0.001 \pm 0.000$ | $0.033 \pm 0.005$ |

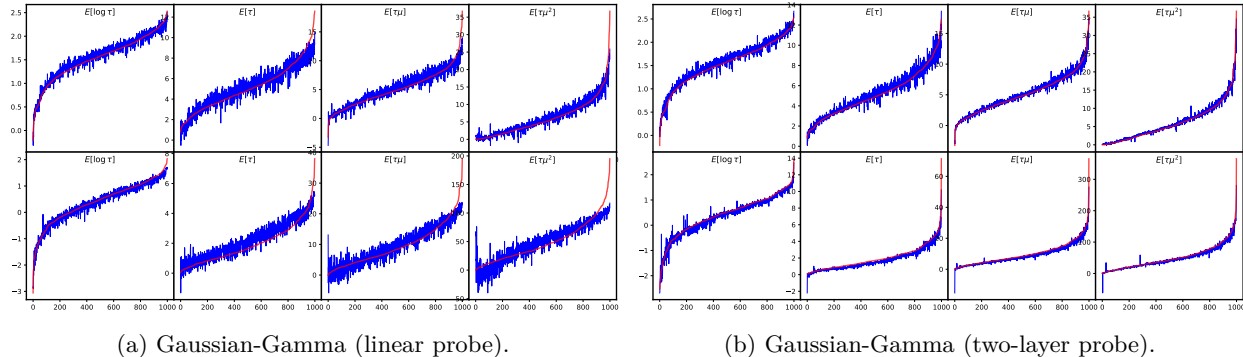

(a) Gaussian-Gamma (linear probe).                    (b) Gaussian-Gamma (two-layer probe).

Figure 8: Probe recovery of transformer-learned posterior distribution moments (blue) and ground truth moments (red) across 1000 test datapoints. The first row shows parameters probed in the general non-OOD case. The second row shows corresponding information in the OOD case.

**Memorization**   Figure 10 conducts the same memorization experiments for the Gamma-Exponential case as in the Gaussian-Gamma case in the main text (Figure 4).

**Parsimonious embeddings**   Table 6 shows results from training an MLP on sufficient statistics to target transformer embeddings. To compute these results, we get mean-squared error, which is then divided by the variance of the transformer embeddings along each dimension separately. The error is averaged across the embedding dimension and subtracted from 1 to yield the mean proportion of variance accounted for in the embedding. These results establish that the relationship runs in both directions: we can decode sufficient statistics from embeddings, and we can predict embeddings from sufficient statistics.

Table 6: Validation variance of transformer embeddings accounted for by training an MLP only on sufficient statistics of training sequences.

| Transformer size | Gaussian-Gamma | Beta-Bernoulli | Gamma-Exponential |
|---|---|---|---|
| 128 | 71.1% | 28.4% | 72.2% |
| 8 | 84.6% | 48.6% | 79.6% |

Figure 11 shows Gaussian-Gamma probing results on sufficient statistics on the 10th token, complementing our exploration of token memorization in transformer (Figure 4).

**Probing results on Wikitext-103**   Table 7 shows probing results on Wikitext-103.

Table 7: Wikitext-103 topic prediction performance based on different LLMs. *On accuracy, the LLAMA2-CHAT performance is not statistically significantly different from that of the MLMs; otherwise, the autoregressive models statistically significantly outperform the non-autoregressive models.*

| Model | Parameters | $K = 20$ | | | $K = 100$ | | |
|---|---|---|---|---|---|---|---|
| | | Accuracy ↑ | L2 loss ↓ | Tot. var. loss ↓ | Accuracy ↑ | L2 loss ↓ | Tot. var. loss ↓ |
| GPT-2 | 124M | $86.7\% \pm 0.5\%$ | $0.025 \pm 0$ | $0.098 \pm 0$ | $73.9\% \pm 2.3\%$ | $0.026 \pm 0.001$ | $0.089 \pm 0.002$ |
| GPT-2-MEDIUM | 355M | $88.2\% \pm 0.6\%$ | $0.024 \pm 0$ | $0.097 \pm 0.001$ | $74.2\% \pm 1.3\%$ | $0.025 \pm 0$ | $0.097 \pm 0.002$ |
| GPT-2-LARGE | 774M | $88.5\% \pm 0.8\%$ | $0.023 \pm 0$ | $0.094 \pm 0.001$ | $74.2\% \pm 1.4\%$ | $0.025 \pm 0$ | $0.088 \pm 0.001$ |
| LLAMA 2 | 7B | $87.3\% \pm 1.7\%$ | $0.023 \pm 0$ | $0.091 \pm 0.001$ | $70.4\% \pm 1.1\%$ | $0.026 \pm 0$ | $0.09 \pm 0.001$ |
| LLAMA 2-CHAT | 7B | $85.3\% \pm 0.7\%$ | $0.024 \pm 0$ | $0.094 \pm 0$ | $69.9\% \pm 1\%$ | $0.026 \pm 0$ | $0.09 \pm 0$ |
| BERT | 110M | $84.9\% \pm 1.1\%$ | $0.027 \pm 0$ | $0.103 \pm 0.001$ | $72.4\% \pm 1.4\%$ | $0.029 \pm 0$ | $0.097 \pm 0.002$ |
| BERT-LARGE | 336M | $85.4\% \pm 1.7\%$ | $0.03 \pm 0$ | $0.111 \pm 0$ | $72.1\% \pm 0.9\%$ | $0.031 \pm 0$ | $0.104 \pm 0$ |
| Null GPT-2 | 124M | $58.1\% \pm 1.8\%$ | $0.121 \pm 0.003$ | $0.247 \pm 0.006$ | $32.9\% \pm 3.2\%$ | $0.099 \pm 0.003$ | $0.195 \pm 0.007$ |

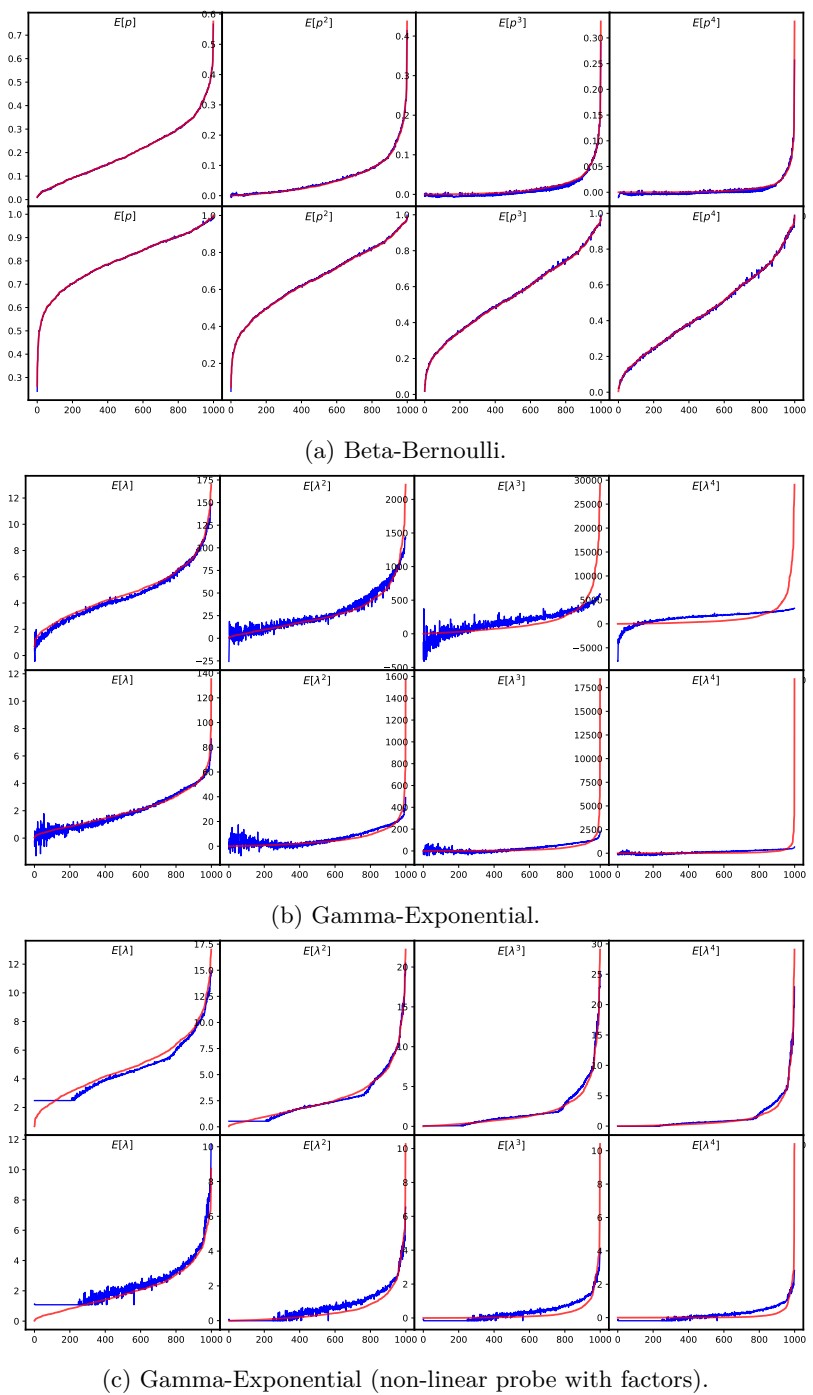

(a) Beta-Bernoulli.

(b) Gamma-Exponential.

(c) Gamma-Exponential (non-linear probe with factors).

Figure 9: Probe recovery of transformer-learned posterior distribution moments (blue) plotted with ground truth moments (red). The first row shows parameters probed on 1000 test datapoints in the general, i.e., non-OOD, case. The second row shows corresponding information in the OOD case.

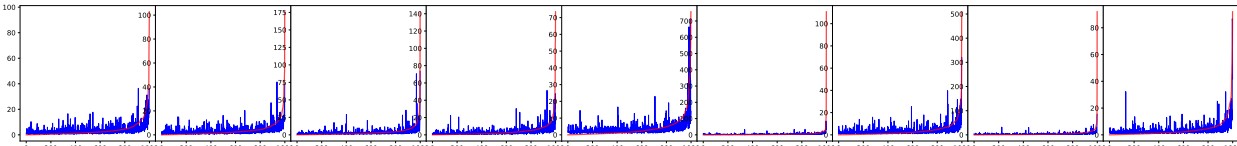

Figure 10: Probing over the first 10 tokens themselves using the 10th token embedding of the transformer. Aside from perfectly encoding the 10th token, this embedding does not show memorization over the other 9 tokens as suggested by the noise in probe recovery. Visually, the noise is less than in the Gaussian-Gamma case (Figure 4). This is partly due to spuriously large magnitude datapoints in this distribution, and the noise here is still significantly higher than in capturing sufficient statistics (Figure 2).

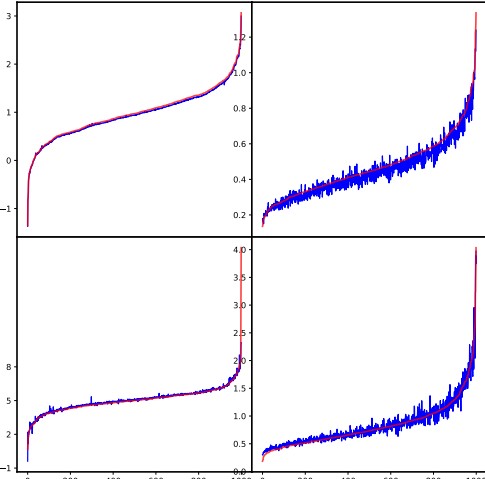

Figure 11: Probing over mean (left) and standard deviation (right) of the first 10 tokens using the 10th token embedding of the transformer in the Gaussian-Gamma dataset. The first row corresponds to the same generation process, and the second row corresponds to the OOD case.

**Inner layer performance on natural corpora**  Table 8 shows topic distribution recovery from LLM inner layers.

Table 8: Across-layer performance in the format of (accuracy, L2-loss), where we move from the word embedder to the final layer (prior to word emission) from left to right.

| Model | Parameters | Layer 0 | Layer 1 | Layer 50% | Layer 75% | Layer 90% | Layer -2 | Layer -1 |
|---|---|---|---|---|---|---|---|---|
| GPT-2 | 124M | $87.4\%, 0.027$ | $87.1\%, 0.025$ | $86.1\%, 0.025$ | $86.7\%, 0.026$ | $86.3\%, 0.026$ | $85.9\%, 0.026$ | $86.7\%, 0.025$ |
| GPT-2-MEDIUM | 355M | $88.2\%, 0.026$ | $86.9\%, 0.024$ | $85.5\%, 0.027$ | $85.1\%, 0.028$ | $83.9\%, 0.03$ | $81.8\%, 0.04$ | $88.2\%, 0.024$ |
| GPT-2-LARGE | 774M | $88.1\%, 0.026$ | $87.6\%, 0.024$ | $86.5\%, 0.025$ | $85.4\%, 0.026$ | $84.5\%, 0.028$ | $82.9\%, 0.03$ | $88.5\%, 0.023$ |
| LLAMA 2 | 7B | $59.2\%, 0.192$ | $74.4\%, 0.104$ | $87.3\%, 0.025$ | $86.6\%, 0.023$ | $86.9\%, 0.024$ | $86.7\%, 0.024$ | $87.3\%, 0.023$ |
| LLAMA 2-CHAT | 7B | $58.8\%, 0.193$ | $74.9\%, 0.098$ | $87.2\%, 0.025$ | $87.5\%, 0.023$ | $86.5\%, 0.024$ | $85.8\%, 0.025$ | $85.3\%, 0.024$ |
| BERT | 110M | $88.4\%, 0.028$ | $88.6\%, 0.024$ | $86.2\%, 0.028$ | $85.9\%, 0.03$ | $86.4\%, 0.028$ | $86.4\%, 0.028$ | $84.9\%, 0.027$ |
| BERT-LARGE | 336M | $88.2\%, 0.029$ | $87.8\%, 0.026$ | $86.4\%, 0.027$ | $85\%, 0.03$ | $85.9\%, 0.029$ | $85.3\%, 0.03$ | $85.4\%, 0.03$ |

**Using only last token embedding on natural corpora**  Using averaged tokens as document embeddings performs better across models in both natural corpora. Here, we also report on results from using only last tokens as document embeddings (Table 13 and Table 14).

Table 9: Autoregressive transformer (AT) and BERT hyperparameters for training on the synthetic datasets.

| Parameter | Tuning range | Chosen value |
|---|---|---|
| Batch-size | $[8, 128]$ | 16 |
| Learning rate | $[3 \cdot 10^{-5}, 10^{-3}]$ | $10^{-4}$ |

Table 10: Probe hyperparameters for training on top of synthetic dataset language models.

| Parameter | Tuning range | Chosen value |
|---|---|---|
| Batch-size | $[8, 64]$ | 16 |
| Learning rate | $[10^{-4}, 0.03]$ | $10^{-3}$ |
| Weight-decay | $[0, 3.4 \cdot 10^{-4}]$ | 0 |
| Embedding choice | {First, Last, Average} | Last for AT / Average for BERT |

Table 11: Probe hyperparameters for training on top of GPT-2, GPT-2-MEDIUM, GPT-2-LARGE, BERT, and BERT-LARGE.

| Parameter | Tuning range | Chosen value |
|---|---|---|
| Batch-size | {128} | 128 |
| Learning rate | $[10^{-5}, 10^{-3}]$ | $3 \cdot 10^{-4}$ |
| Weight-decay | $[0, 3.4]$ | $3.4 \cdot 10^{-3}$ |
| Embedding choice | {First, Last, Average} | Average |

Table 12: Probe hyperparameters for training on top of LLAMA 2 and LLAMA 2-CHAT.

| Parameter | Tuning range | Chosen value |
|---|---|---|
| Batch-size | {128} | 128 |
| Learning rate | $[10^{-5}, 10^{-3}]$ | $10^{-4}$ |
| Weight-decay | $[0, 3.4]$ | 0.34 |
| Embedding choice | {First, Last, Average} | Average |

Table 13: 20NG topic prediction performance based on different LLMs using the last token as document embedding.

| Model | Parameters | $K = 20$ | | | $K = 100$ | | |
|---|---|---|---|---|---|---|---|
| | | Accuracy ↑ | L2 loss ↓ | Tot. var. loss ↓ | Accuracy ↑ | L2 loss ↓ | Tot. var. loss ↓ |
| GPT-2 | 124M | $47.8\% \pm 1.8\%$ | $0.143 \pm 0.001$ | $0.26 \pm 0.002$ | $31\% \pm 3\%$ | $0.115 \pm 00.001$ | $0.217 \pm 0.002$ |
| GPT-2-MEDIUM | 355M | $47.6\% \pm 1.2\%$ | $0.144 \pm 0.001$ | $0.26 \pm 0.001$ | $30.8\% \pm 3\%$ | $0.116 \pm 0.001$ | $0.217 \pm 0.002$ |
| GPT-2-LARGE | 774M | $47.8\% \pm 1.8\%$ | $0.144 \pm 0$ | $0.261 \pm 0.002$ | $31.1\% \pm 2.9\%$ | $0.116 \pm 0$ | $0.217 \pm 0.002$ |
| LLAMA 2 | 7B | $45.4\% \pm 2\%$ | $0.152 \pm 0$ | $0.267 \pm 0.001$ | $27.9\% \pm 2.9\%$ | $0.122 \pm 0.001$ | $0.223 \pm 0.003$ |
| LLAMA 2-CHAT | 7B | $46.1\% \pm 1.8\%$ | $0.15 \pm 0.001$ | $0.265 \pm 0.001$ | $28.5\% \pm 2.5\%$ | $0.122 \pm 0.001$ | $0.222 \pm 0.003$ |
| BERT | 110M | $51.7\% \pm 1.4\%$ | $0.126 \pm 0.002$ | $0.236 \pm 0.002$ | $34.9\% \pm 2.8\%$ | $0.108 \pm 0$ | $0.204 \pm 0.001$ |
| BERT-LARGE | 336M | $52.2\% \pm 2.1\%$ | $0.125 \pm 0.001$ | $0.235 \pm 0.002$ | $34.7\% \pm 2.7\%$ | $0.107 \pm 0.001$ | $0.201 \pm 0.001$ |

Table 14: WikiText-103 topic prediction performance based on different LLMs using the last token as document embedding.

| Model | Parameters | $K = 20$ | | | $K = 100$ | | |
|---|---|---|---|---|---|---|---|
| | | Accuracy ↑ | L2 loss ↓ | Tot. var. loss ↓ | Accuracy ↑ | L2 loss ↓ | Tot. var. loss ↓ |
| GPT-2 | 124M | $69.2\% \pm 1.4\%$ | $0.079 \pm 0.003$ | $0.191 \pm 0.005$ | $50.9\% \pm 1.3\%$ | $0.055 \pm 0.001$ | $0.146 \pm 0.003$ |
| GPT-2-MEDIUM | 355M | $70.7\% \pm 0.6\%$ | $0.077 \pm 0.002$ | $0.188 \pm 0.004$ | $51.6\% \pm 1.9\%$ | $0.056 \pm 0.001$ | $0.146 \pm 0.003$ |
| GPT-2-LARGE | 774M | $72\% \pm 0.6\%$ | $0.074 \pm 0.001$ | $0.184 \pm 0.003$ | $52.8\% \pm 2\%$ | $0.053 \pm 0.001$ | $0.143 \pm 0.002$ |
| LLAMA 2 | 7B | $69.7\% \pm 2.3\%$ | $0.075 \pm 0.002$ | $0.187 \pm 0.003$ | $51.9\% \pm 1.4\%$ | $0.054 \pm 0.001$ | $0.144 \pm 0.003$ |
| LLAMA 2-CHAT | 7B | $71\% \pm 1.8\%$ | $0.073 \pm 0.001$ | $0.181 \pm 0.002$ | $54.2\% \pm 0.8\%$ | $0.052 \pm 0.002$ | $0.141 \pm 0.005$ |
| BERT | 110M | $73.7\% \pm 1.8\%$ | $0.067 \pm 0$ | $0.177 \pm 0.002$ | $58.9\% \pm 0.9\%$ | $0.05 \pm 0.002$ | $0.14 \pm 0.001$ |
| BERT-LARGE | 336M | $63.9\% \pm 3\%$ | $0.1 \pm 0.001$ | $0.219 \pm 0.001$ | $50\% \pm 1\%$ | $0.065 \pm 0.002$ | $0.164 \pm 0.003$ |

