# OpenReview forum: "What Should Embeddings Embed? Autoregressive Models Represent Latent Generating Distributions"
_TMLR — Accepted by TMLR_

### Review · Reviewer_iADS · 2025-05-12

**Summary Of Contributions:**

The main contribution of this paper is to draw the connection between the activations from autoregressive language models and the concept of predictive sufficient statistics, and to validate this through empirical studies on Transformer-based models. The authors point out that if a language model captures the predictive distribution $p(x_{n+1} | x_{1:n})$ perfectly, then it must have learned to represent the predictive sufficient statistics of the data distribution. To test to what extent learning to represent sufficient statistics occurs in these models in practice, they focus on data generated from two families of sequential probabilistic models containing instances with known and tractable sufficient statistics: exchangeable models and latent state models. Their methodology for recovering the sufficient statistics from the trained autoregressive LM is to train a linear probe on the last layer embedding.

The empirical studies show the following: the sufficient statistics can be recovered in simple synthetic settings for exchangeable and latent state models, and that as the main experimental context, topic modeling, transitions from easier (still synthetic) settings to natural text corpora, a significant amount of the latent topic mixture can still be recovered.

**Audience:**

Yes

**Claims And Evidence:**

Yes

**Requested Changes:**

**Minor**: In Section 4.1.2, did you find similar results on memorization for the other studied models (Beta-Bernoulli and Gamma-Exponential)? You could consider adding this to the appendix.

**Recommended**: Can you add some discussion on how the BERT topic model in Section 4.3 fits with the Section 3 framework, and additional interpretation of the BERT results in this light? Section 3 focuses on next-word prediction, whereas BERT is a masked language model (i.e., it predicts the current word), and is not autoregressive.

**Recommended**: The conclusion could benefit from more discussion on the intended takeaways — additionally, while the experiments focus heavily on topic models, the paper also hints that the findings are more broadly applicable (such as in the intriguing last paragraph of the introduction). It would be interesting to revisit this in the conclusion, with connections to the empirical results.

**Strengths And Weaknesses:**

**Strengths**:
- The experiments are extensive and convincing. The authors thought carefully about appropriate controls to ensure validity of the results (controlling for the probe's performance, as well as the two baselines: the end-to-end word embedder and Null GPT-2). The findings of this paper are relevant to TMLR audience members interested in the connection between LLMs and Bayesian inference, as well as those involved in the field of mechanistic interpretability.

**Weaknesses**:
- While the experiments are thorough, a large part of the paper’s 12 pages is devoted to explaining the experiments’ setup/details, and I feel that the paper would benefit from a more organized discussion of the findings. A suggestion is to move Section 4.2 to the appendix (the results on HMMs are interesting, but similar to the exchangeable results) and to expand Section 4.4.
- The organization and polish of the experiments section could be improved. There are additional changes that could be made to improve readability, such as in the beginning of Section 4.1.2, implementation details on the OOD experiment are provided before the experiment itself is described. I also find some of the tables a little hard to read due to formatting (e.g., Table 3).
- I have provided some open questions, changes, and suggestions for the authors to address.

**Questions**:
- For the topic model experiments, I am not sure about the justification for why you chose to probe only the last layer embedding. For example, Thompson and Mimno (2020) [1] find that using GPT-2’s final layer as the representation for a clustering-based topic model performs very poorly, with significant improvement from using either the $(L-1)$th or $(L-2)$th layer. The theory outlined in Section 3 would still hold if the sufficient statistics were present in earlier, but not later, layers.
- Section 4.1.2: how important is ruling out memorization of the inputs in the last layer embedding to the claims of the paper? Would it be possible for the model to use memorized inputs to reconstruct the predictive distribution in the last linear layer?
- Table 3: The performance of the true model (LDA) in this experiment seems a little low, do you have an idea of why that is?

**Stylistic suggestions:**
- While Figure 1 shows three generative processes, the discrete hypothesis model is a special case of the exchangeable data model, as you describe in Section 3.1.1. It would be more clear if this relationship was shown in the figure. Also, in the sentence “The relevant predictive sufficient statistics are the sufficient statistics for $\theta$”, drawing the parallel between the sufficient statistic and the posterior would be a bit more clear if you listed the latent variables in question, ”$\theta$, $z_{n+1}$, $h$”.
- Section 4.1.1: I suggest putting the full generative process for the Gaussian-Gamma model in the main text, as a reader who doesn’t know this model wouldn’t be able to infer what it is from just the section text and equation. You might also want to say something like “We describe the Gaussian-Gamma model as an example of a Bayesian conjugate model, with full details on the remaining two in Appendix A.1.”
- Section 4.1.2: The OOD abbreviation should be defined where it first appears in the text.
- Section 4.1.2: What do you mean by “suggesting a similar robustness with a Bayes-optimal agent”?
- Section 4.1.2: It would be more clear to use the mathematical formula for the captured variance in embeddings instead of writing it in words.
- Section 4.2.1: typo: “hyperparamters”
- Section 4.2.2: missing word: “should encode the predictive sufficient [statistics]”
- Table 2: In the caption, do you mean standard deviation or standard error?
- Section 4.3: typo: “that draws draws words”
- Section 4.3.2: I understood the general idea of this section, but found the explanation of the experiment details confusing. How do the five distinct topic models differ (e.g., can you provide hyperparameters)? Can you use some mathematical notation to explain how the probe training vs. prediction works?
- Section 4.4.2: “20NG can be naturally grouped into twenty topics” — you should specify that this is because each post is from a newsgroup about a specified subject.
- Section 4.4.2: “LDA-learned topic mixtures with $K = 20, 100$” $\to$ “with $K \in ${$20, 100$}”

[1] Topic Modeling with Contextualized Word Representation Clusters (Thompson and Mimno, 2020) (https://arxiv.org/abs/2010.12626)

---

> ### Author Response · Authors · 2025-06-04
> **Author Response to Reviewer iADS (part 1)**
>
> We thank the reviewer for the careful feedback, which will improve the paper. Below we address the major concerns. For the stylistic suggestions and requested changes, we will incorporate them in the revised paper.
>
> **Q1.** Last layer vs. the L-1 th or L-2 th layer.
>
> **Response 1.** Thanks for the helpful discussion. The L-1th layer in the topic model case has $81.9\% \pm 1.1\%$ accuracy, compared to $82.8\% \pm 0.5\%$ of the last layer (in the paper). The problem of which layer is expected to perform better may depend on the generative process. A hypothesis is that if the latent variable is in an elementary stage of the generative process, and the likelihood function given this latent variable is complex, then this latent variable may be better encoded by the earlier layers; on the contrary, if the data generation given the targeted latent variable is simple, then later layers have the advantage.
>
> **Q2.** Section 4.1.2: rationale for experiments on memorization.
>
> **Response 2.** It would be possible for the model to use memorized inputs to reconstruct the predictive distribution, so indeed even if there were memorization, it wouldn’t immediately refute the claims of the paper. However, a concern is that the probe may be doing the job of finding the latent variables by using the transformer’s memorized inputs, so ruling out memorization strengthens our claim by suggesting that this information is in the transformer. We will clarify in this subsection.
>
> **Q3.** Table 3: The performance of the true model (LDA) in this experiment seems a little low.
>
> **Response 3.** We hypothesize that randomness in the generation process of the original LDA leads to an upper bound in the LDA that tries to recover this process. That said, the results do confirm that LDA performs best among all models that try to recover the generative process.
>
> **Weakness 1 & 2.** Adjustments to organization regarding experimental setup and details.
>
> **Response 1.** Thank you for your suggestion. We will adjust the organization regarding experimental setup and details accordingly.
>
> **Requested Change 1.** Memorization for the other studied models.
>
> **Response 1.** We conducted these memorization studies on the Gamma-Exponential case, where we observe a similarly noisier recovery on the previous tokens, compared to sufficient stats. We will add this result in the revision.
>
> **Requested Change 2.** Whether BERT fits with the Section 3 framework, and additional interpretation of the BERT results.
>
> **Response 2.** We find that the masked objective can also be broken down into implicit Bayesian inference like the autoregressive one, but it differs from autoregressive language models in a way that results in less expressivity. The log objective can be extended as follows,
> $\sum_{n \in M} \log \int p(x_n | \theta)p(\theta|x_{i,i\in U})d\theta$, where $M$ denotes the set of masked indices, and $U$ denotes the set of unmasked indices. This can be shown by a simple extension of the derivation for the autoregressive version, and we will add this to the appendix in the revised paper.
>
> The difference between the two objectives is that in the summation, the prediction of each token $x_n$ uses the same posterior over the latent variable $p(\theta|x_{i,i\in U})$. In other words, each token $x_n$ is predicted independently from the latent variable $\theta$. As a result, MLM forms a less expressive Bayesian inference objective than autoregressive models. This can be connected to recovering topic mixtures in Table 3, where Bert performance is notably worse than that of an autoregressive transformer.
>
> **Requested Change 3.** Intended takeaways and broader applications.
>
> **Response 3.** We will add a discussion section that begins with a clearer conclusion on takeaways. We will also discuss broader implications. Here is a summary of our intended addition:
>
> Our results help inform construction and use of embeddings in practice, and can be applied to generative models that satisfy the conditions discussed by the paper. The analyses can be naturally extended to other latent variables that do not depend heavily on word order, such as the author type of the document [Andreas, 2022] or the author’s sentiment [Radford et al., 2017].
>
> As another example, our results can inform time series modelling. A practitioner might have a time series dataset with underlying factors that are informed by human experts. If they want to use deep autoregressive models to construct an embedding that contains a certain kind of information from the input, they may need to ensure that this information is a predictive sufficient statistic for the task.

---

> > ### Author Response · Authors · 2025-06-04
> > **References**
> >
> > Jacob Andreas. Language models as agent models. In Findings of the Association for Computational Linguistics: EMNLP 2022, pages 5769–5779, Abu Dhabi, United Arab Emirates, December 2022. Association for Computational Linguistics.
> >
> > Alec Radford, Rafal Jozefowicz, and Ilya Sutskever. Learning to generate reviews and discovering sentiment. arXiv preprint arXiv:1704.01444, 2017.

---

> > ### Comment · Reviewer_iADS · 2025-06-20
> > **Thank you for your response**
> >
> > Thanks to the authors for their detailed response to the questions brought up in my review! After reading the other reviews and responses and checking the revised paper, I believe my concerns have been adequately addressed. This work is a nice contribution connecting Bayesian inference and LLM interpretability.

---

### Review · Reviewer_DJUX · 2025-05-14

**Summary Of Contributions:**

In this paper, the authors investigate what embeddings represent in autoregressive language models. The authors demonstrate that these embeddings can be viewed as predictive sufficient statistics, capturing key latent structures underlying text generation. By examining interpretable scenarios such as exchangeable models and latent state models, they empirically show that transformer embeddings encode meaningful latent distributions, generalize well to out-of-distribution data. Extending their analysis to more realistic settings, the authors specifically focus on topic modeling, revealing that large language models implicitly encode topic mixtures similar to those inferred by LDA. Experiments are conducted on both synthetic adn real datasets. Overall, this paper studies an interesting topic and is well-written, the experiments are also well-designed.

**Audience:**

Yes

**Broader Impact Concerns:**

I have no concerns about the impact

**Claims And Evidence:**

Yes

**Requested Changes:**

See above comments

**Strengths And Weaknesses:**

Pros:

[1] The paper is well-written and easy to follow. The figures are well-designed.

[2] I like the way to do experiments with both synthetic data and real data. Sometimes with synthetic data, it is easier for controlled experiments.

[3] This paper focuses on an important and interesting problem, which will benefit the LLM community significantly.

Cons:

While the paper is well-written, some additional figures and concrete examples might further improve the readability to the general audience in the community. For example, what topics are encoded by different layers of LLMs.

In Section 4.4.2,  “we observe that Llama-2 relies on its inner layers to encode topic mixtures.” Is there a potential to explain why there is such phenomenon?

There are multiple interesting findings in this paper. I would like to suggest adding a section or subsection, to summarize the Lessons Learned or Observations with bullet forms. This will benefit the readers and future studies in this area.

Could you also discuss the limitations and potential future directions?

While I aggree with the importance of this work, could you also discuss what potential implications or what applications could benefit from understanding the embeddings? For example, could we train a better LLM or LLM could be explainable.

Could you also discuss the relations to studies about embedding understanding in image data (e.g., R1), as well as embedding interpretation in other language models (e.g., R2)?

Typos: In section 4.3, the first sentence might contain some repeated words, e.g., “draw”.

R1: Exploiting Interpretable Capabilities with Concept-Enhanced Diffusion and Prototype Networks, 2024

R2: Variational Language Concepts for Interpreting Foundation Language Models, 2024

---

> ### Author Response · Authors · 2025-06-04
> **Author Response to Reviewer DJUX**
>
> We thank the reviewer for helpful feedback, which will improve the paper. Below we address the main points raised by the reviewer. We will also upload a revised paper that addresses these points.
>
> **Q1.** Additional figures and concrete examples.
>
> **Response 1.** Thank you for making this suggestion. In the revised paper, we will add examples where we show an example passage together with what topic mixtures are predicted by the probes and by LDA.
>
> **Q2.** In Section 4.4.2, “we observe that Llama-2 relies on its inner layers to encode topic mixtures.” Is there a potential to explain why there is such phenomenon?
>
> **Response 2.** A possible explanation is that the size of Llama-2 allows it to better perform autoregressive modelling, which then leads to better encoding of latent generating distributions (as reflected in Figure 5c) (the context here is that Llama-2 performance-by-layer improves progressively, with the word embedding layer and earlier layers performing worse). We have added a note about this possibility to the paper.
>
> **Q3.** Adding Lessons Learned or Observations.
>
> **Response 3.** Thank you for pointing this out. We will add a discussion section in the revised paper and start this section with a clearer conclusion on lessons learned.
>
> **Q4.** Limitations and potential future directions.
>
> **Response 4.** We will add this to our discussion section in the revised paper. It is possible to extend our analyses to deep learning models for other modalities. Also, a study on the relationship between task difficulty and the transformer’s ability to embed sufficient statistics would be helpful. As seen in the hypothesis space experiments, predictive sufficient statistics become harder to decode as task difficulty increases.
>
> **Q5.** potential implications or applications.
>
> **Response 5.** We believe that our results are useful in identifying how to construct and use embeddings in practice. Specifically, they provide limits on what embeddings should be expected to contain that can inform experiments that practitioners run. Namely, a practitioner might have a sequential dataset (e.g. time series data) with underlying factors that are informed by human experts. If the practitioner wants to use these deep autoregressive models to construct an embedding that contains a certain kind of statistical information from the input, they may need to ensure that this information is a predictive sufficient statistic for the task that is given to the model. We will add this to the discussion section in the revision.
>
> **Q6.** Could you also discuss the relations to studies about embedding understanding in image data (e.g., R1), as well as embedding interpretation in other language models (e.g., R2)?
>
> **Response 6.** Thank you for pointing out this area of discussion. R1 discusses novel architectures that perform image concept classification where some of their intermediate embeddings are forced to be more interpretable. In language models, there have also been proposals for architectures that are more interpretable  (e.g. Hewitt et al. 2023). However, scalable architectures have been dominated by the less interpretable transformers, and the aim of our paper is to understand if and why concepts are still embedded in these models. Nonetheless, building more inherently interpretable models is an interesting direction for future work, and our results could inform such efforts by providing insights into what sorts of information should be encoded; we will include this possibility in the Discussion section that we add about practical implications (see our response to Q6).
>
> R2 provides a distinct way to extract from language models concepts that come in the form of a mixed membership model (including the topic model). Our focus is more explanatory as we analyze why concepts that come in the form of sufficient stats should be embedded by autoregressive language models (and our methodology of probing allows us to study latent variables beyond topic mixtures).
>
> We will include these related works in our Related Work and/or Discussion sections.
>
> **Q7.** Typos: In section 4.3, the first sentence might contain some repeated words, e.g., “draw”.
>
> **Response 7.** Thank you for pointing this out. It will be corrected in the revised paper.
>
> **Reference**
>
> John Hewitt, John Thickstun, Christopher Manning, and Percy Liang. Backpack Language Models. Proceedings of the Association for Computational Linguistics (2023).

---

### Review · Reviewer_a9NK · 2025-05-23

**Summary Of Contributions:**

This paper provides a principled theoretical and empirical investigation into what embeddings in autoregressive language models (LMs) represent. The authors argue that these embeddings serve as predictive sufficient statistics—compressing all the necessary information from the past to predict the next token. They ground their claims in two canonical data-generation processes: exchangeable models (e.g., topic models like LDA) and latent state models (e.g., HMMs). Theoretical analyses are supported by probing studies on synthetic and real-world datasets. The paper shows that embeddings capture latent generating distributions, generalize out-of-distribution, and avoid trivial token memorization. The authors further validate their framework on LDA-generated data, mixed HMM-LDA synthetic data, and natural corpora (20 Newsgroups and WikiText-103) using pretrained LLMs (GPT-2, LLaMA-2, BERT).

**Audience:**

Yes

**Claims And Evidence:**

Yes

**Requested Changes:**

1. Please clarify the memorization experiment shown in Table 4 and the last paragraph of page 7. To me, it is not clear what the authors are trying to show.

2. Please clarify/strengthen the last real-world experiment. Please see weaknesses for detailed comments.

3. Many figures' x and y axes are not clearly labeled, and the ticks are too small.

**Strengths And Weaknesses:**

Strengths:

1. The paper provides a principled Bayesian framing of what information the embeddings of LMs should contain: sufficient statistics. Theoretically, this is also the only information that the embedding should contain.

2. The authors analyze diverse data generation models, including exchangeable models, LDA, and HMMs, and also provide probing-based empirical evidence for each case.

3. The authors also provide some ablation studies to show that observed effects are likely due to the LM’s representations rather than overly powerful probes.

4. The hypothesis is verified across different model architectures used in the experiments: BERT, GPT2, LLAMA 2.

5. The authors also attempt to extend the hypothesis to real-world text data.

Weaknesses:

1. The paper's theoretical and empirical results focus on simple data generative models, which are quite far from the real-world text data distribution. A more sophisticated data distribution model that takes syntax and semantics into consideration would greatly strengthen the paper.

2. The memorization defined in the last paragraph of page 7 is a bit weird. It is only natural that the embedding of the last token does not memorize the previous tokens, as it is supposed to predict the next token. In my opinion, memorization should represent whether the LM is memorizing the training data or learning the distribution/sufficient statistics.

3. The experiment with real-world text data (20NG) is a bit of a far stretch. The assumption is still that 20NG follows an LDA generative process, which clearly is not. It is only natural that a trained LM can catch the semantics of a piece of text, thus being predictive of its topic, compared to a randomly initialized untrained baseline. I don't see how the sufficient statistics hypothesis is supported through this experiment.

4. It would be nice if there were some real-world implications of the conclusion: would training LM embeddings to better encode the sufficient statistics improve LM performance? The real-world LMs likely do not perfectly learn the autoregressive factorization of the underlying data distribution.

---

> ### Author Response · Authors · 2025-06-04
> **Author Response to Reviewer a9NK**
>
> We thank the reviewer for the helpful feedback, which will improve the paper. Below we address the main concerns. We will also include the requested changes in the revised paper.
>
> **Weakness 1.** More sophisticated data distribution model that takes syntax and semantics into consideration.
>
> **Response 1.** We agree that more complex data distributions would be interesting. That said, our Section 4.4.1 does explore a more sophisticated generative model that considers both syntax and semantics, where the data is generated by a mix of HMM (representing syntax) and LDA (representing semantics). Though these models are simple in their formal structure, both have proven surprisingly effective at modeling their respective aspects of natural language: HMMs are effective at capturing much of syntax as shown by their strong performance at syntactic tagging tasks in natural corpora (e.g., part-of-speech tagging), and LDA is effective for topic modeling (a semantic task) on natural corpora. Thus, we view these two models as reasonable first approximations of syntax and semantics as they appear in real-world text distributions.
>
> **Weakness 2.** Rationale behind memorization experiments.
>
> **Response 2.** It’s true that the embedding should predict the next token, but the question is how it does so: it could do so using sufficient statistics (as we hypothesize), but in principle it could instead simply memorize previous tokens, since the previous tokens do provide enough information to predict the next token. These two possibilities create a possible confound behind probing: The success of the probe could either be explained by the transformer modeling the sufficient stats (which the probe can then read out), or by the transformer memorizing previous inputs which the probe then uses to infer these sufficient stats by itself, without that information existing in the transformer. Showing little to no memorization would rule out the second possibility. We will clarify this point in this subsection in 4.1.2.
>
> Thank you also for pointing out this ambiguity in our terminology: We agree that “memorization” is more often used to talk about memorizing training data, whereas we use it to mean memorizing the previous context tokens during inference time. To clarify what we mean, we will revise the paragraph heading from “Ruling out memorization” to “Memorizing the context or storing sufficient statistics of the context?”, and we will also revise the paragraph along similar lines.
>
> **Weakness 3.** 20NG does not follow LDA, and why is the sufficient statistics hypothesis supported through this experiment.
>
> **Response 3.** Although topics by themselves do not model real-world text, it is reasonable to assume that they form a sub-component of language, and are a factor among numerous other factors involved in generating text (for reference, the first paragraph of Section 4.4.1 discusses this point in more detail). While other experiments more directly support our sufficient stats hypothesis, for real world text-data, our intent is to show how it can also be explanatory for why semantics can be encoded by transformers modelling such data. Additionally, our design of decoding topic mixtures from single token residual streams is distinct from previous work and confirms our explanatory hypothesis. We will clarify this point in the beginning of Section 4.4.2.

---

> > ### Author Response · Authors · 2025-06-04
> > **Author Response to Reviewer a9NK (part 2)**
> >
> > **Weakness 4.** Real world implications / real-world LMs likely do not perfectly learn the autoregressive factorization of the underlying data distribution.
> >
> > **Response 4.** Thank you for pointing out this area for discussion. We have added a discussion section in the revised paper engaging with this point. We did discover that better autoregressive modeling leads to better learning of the latent data distribution factorization (last paragraph of Section 4.4.1). Directly targeting sufficient stats, which the reviewer mentions, is to some extent done by neural-augmented topic models (Dieng, et al. (2019), but explicitly training a language model that embeds more factors would be challenging as it is difficult to identify an adequate list of sufficient stats behind real world data. That said, related work such as McCoy et al. (2025) discovers that pre-training on synthetic syntactic data helps later training on real text, and we will include this discussion in the revision.
> >
> > One other practical implication of the conclusion regards future directions for interpretability. Understanding the inner workings of LLMs is important for AI safety and trustworthiness. However, the sheer size of LLMs makes it challenging to analyze them. Our results suggest that interpretable models of document structure - such as topic models - can provide useful guidance about what mechanistic interpretability should look for, giving us a starting point for the daunting task of analyzing LLMs. This in turn means that one goal for interpretability work should be enhancing models like LDA because such advances will in turn sharpen our ability to interpret LLM representations.
> >
> > **Requested Change 1.** Clarify memorization experiments.
> >
> > **Response 1.** See our response to Weakness 2.
> >
> > **Requested Change 2.** Please clarify/strengthen the last real-world experiment. Please see weaknesses for detailed comments.
> >
> > **Response 2.** See our response to Weakness 3 & 4.
> >
> > **Requested Change 3.** Figure axes and ticks.
> >
> > **Response 3.** Thank you. We will modify the figures accordingly.
> >
> > **Reference**
> >
> > McCoy, R.T., Griffiths, T.L. Modeling rapid language learning by distilling Bayesian priors into artificial neural networks. Nat Commun 16, 4676 (2025).
> >
> > Dieng, A.B., Ruiz, F.J., & Blei, D.M. (2019). The Dynamic Embedded Topic Model. ArXiv, abs/1907.05545.

---

### Decision · Action_Editor_3E7o · 2025-06-22

**Recommendation:** Accept as is

**Additional Comments:**

This paper presents a clear and compelling investigation into what information is captured by the embeddings of autoregressive models. The core contribution is the elegant connection drawn between the autoregressive objective and the learning of predictive sufficient statistics. This theoretical insight is seamlessly combined with a series of rigorous experiments.

The reviewers were unanimously in favor of acceptance. All their concerns were addressed by the authors during the discussion period. This work sheds some light on the interpretation of the internal representations of large language models.

**Audience:**

Yes

**Audience Explanation:**

This paper may be of interest to researchers and practitioners in large language models, deep learning and Bayesian inference. Scholars working specifically in LLM interpretability will definitely benefit from reading this paper.

**Claims And Evidence:**

Yes

**Claims Explanation:**

The claims of the paper are well-supported, with a principled theoretical framework and extensive empirical evidence.

In particular, the paper argues that the embeddings of autoregressive language models learn to represent predictive sufficient statistics of the underlying data generating distribution. The authors ground this claim in Bayesian theory, and analyze two major classes of generative processes: exchangeable models and latent state models.

This theoretical foundation is backed by a comprehensive suite of experiments that the reviewers found convincing:
- Synthetic data from simple exchangeable models (e.g., Gaussian-Gamma or Beta-Bernoulli) and latent state models (HMMs), where the ground-truth sufficient statistics are known and can be directly compared against.
- Topic models (LDA), extending the analysis to a more complex exchangeable model relevant to text.
- A hybrid HMM-LDA model, which begins to approximate the mixture of syntactic and semantic structures found in natural language.

The authors use multiple language model architectures (GPT-2, LLaMA 2, and BERT). The reviewers pointed out that the generative models used in the synthetic experiments are arguably simplifications of the true complexity of language. Despite that, the evidence presented in the paper is clear, accurate, and supports the paper's central claims.